# Targeting Insulin Resistance, Reactive Oxygen Species, Inflammation, Programmed Cell Death, ER Stress, and Mitochondrial Dysfunction for the Therapeutic Prevention of Free Fatty Acid-Induced Vascular Endothelial Lipotoxicity

**DOI:** 10.3390/antiox13121486

**Published:** 2024-12-05

**Authors:** Chong-Sun Khoi, Tzu-Yu Lin, Chih-Kang Chiang

**Affiliations:** 1Department of Anesthesiology, Far-Eastern Memorial Hospital, New Taipei City 220216, Taiwan; d05447005@ntu.edu.tw; 2Graduate School of Biotechnology and Bioengineering, College of Engineering, Yuan Ze University, Taoyuan City 320315, Taiwan; 3Department of Mechanical Engineering, College of Engineering, Yuan Ze University, Taoyuan City 320315, Taiwan; 4Graduate Institute of Toxicology, College of Medicine, National Taiwan University, Taipei City 10617, Taiwan; 5Department of Internal Medicine, College of Medicine, National Taiwan University, Taipei City 10617, Taiwan; 6Department of Integrated Diagnostics & Therapeutics, National Taiwan University Hospital, Taipei City 100229, Taiwan

**Keywords:** fatty acids, palmitic acid, endothelial dysfunction, prevention

## Abstract

Excessive intake of free fatty acids (FFAs), especially saturated fatty acids, can lead to atherosclerosis and increase the incidence of cardiovascular diseases. FFAs also contribute to obesity, hyperlipidemia, and nonalcoholic fatty liver disease. Palmitic acid (PA) is human plasma’s most abundant saturated fatty acid. It is often used to study the toxicity caused by free fatty acids in different organs, including vascular lipotoxicity. Fatty acid overload induces endothelial dysfunction through various molecular mechanisms. Endothelial dysfunction alters vascular homeostasis by reducing vasodilation and increasing proinflammatory and prothrombotic states. It is also linked to atherosclerosis, which leads to coronary artery disease, peripheral artery disease, and stroke. In this review, we summarize the latest studies, revealing the molecular mechanism of free fatty acid-induced vascular dysfunction, targeting insulin resistance, reactive oxygen species, inflammation, programmed cell death, ER stress, and mitochondrial dysfunction. Meanwhile, this review provides new strategies and perspectives for preventing and reducing the impact of cardiovascular diseases on human health through the relevant targeting molecular mechanism.

## 1. Introduction

Fatty acids (FAs) belong to carboxylic acids and are classified into saturated fatty acids (SFA) with no carbon–carbon bonds and unsaturated fatty acids (UFAs) with one or more carbon–carbon bonds. In addition, fatty acids are also classified into short-chain fatty acids, medium-chain fatty acids, and long-chain fatty acids according to their tail length [1]. 

FAs derive from dietary animal fat, plant oils, fish oil, nuts, fruit [2], or carbohydrates in adipose tissue, liver, and mammary gland. FAs contained in dietary fat as triglycerides (TG) are digested by lipase to form FAs, which are adsorbed by intestinal villi to form chylomicrons, transported by the lymphatic system, and finally enter the blood circulation [3]. Triglycerides, phospholipids, and cholesterol esters are FA that exist in the body as esters. Free fatty acids (FFAs) are circulating plasma FAs that are not present in ester types, which are usually bound to albumin.

Ingest excessive FFA are the main cause of obesity, hyperlipidemia, and nonalcoholic fatty liver disease [4]. In addition, a high level of FFAs contributes to atherosclerotic formation [5]. An Australian cohort study using the Blue Mountains Eye Study (BMES1) data collected fatty acid intake and food sources in Australian people older than 55 years old. Among the daily energy intake of total fat, SFA was the highest portion at 12.2%, monounsaturated fatty acid (MUFA) was 11.2%, and polyunsaturated fatty acid (PUFA) was 5.0%, respectively. Palmitic acid (PA) accounts for the most portions of SFA intake, oleic acid was the main portion of MUFA intake, and linoleic acid was the main portion of total PUFA intake. Milk and milk products mainly contributed to SFA, meat and meat products mainly contributed to MUFA intake, and fats and oils mainly contributed to PUFA. This study found that daily intake of SFA as 12.2% of energy was higher than Australian recommendations, and it provided basic analysis for elucidating the incidence of chronic disease and fatty acids [6].

A Korean study using the Korean National Health and Nutrition Examination Survey (KNHNES) to investigate fatty acid intake and obesity prevalence from 2007 to 2017 revealed that elevated obesity prevalence from 2015 to 2017 showed a significant increase in BMI in the obesity group in both genders, and fat intake has increased in both genders. Intake of total fatty acid, including saturated fatty acids (SFAs), monounsaturated fatty acids (MUFAs), and polyunsaturated fatty acids (PUFAs), has increased in both men and women. This study found that animal foods are the highest fat intake source among seven food groups; more than 50% of consumed SFA and MUFA mainly come from animal-originated products like beef, pork, chicken, milk, milk products, eggs, bread, and cookies. This study implicated that obesity may correlate with increasing fat intake, and intake of N-3 fatty acids should be increased [7].

Another takeaway food content SFA and trans-fatty acid (TFA) study from the UK demonstrated that the highest SFA was in English and pizza meal categories with a 35.7 g/portion median average; kebab and chips have the highest SFA with a 47.6 g/portion median amount. Regular intake of these foods may increase the risk of coronary artery disease (CAD), type II DM, and obesity due to excessively high SFAs in these foods [8]. Reducing the intake of dietary SFA significantly reduced total cholesterol, LDL, and diastolic pressure in children and adolescents aged between 2 and 19. These findings implied that reducing the consumption of fried fast foods, snacks, processed meats, and fatty meats would decrease intake of SFA to prevent later CVD in life [9].

Endothelial cells constitute a continuous single layer of the endothelium barrier that modulates vascular homeostasis, development, and pathogenesis [10]. Impairment of endothelial function is involved in increased activation of endothelial cells and the beginning of endothelial dysfunction [11]. Endothelial dysfunction is associated with coronary artery disease, peripheral artery disease, and chronic heart failure, which is characterized by reduced vasodilation and increased proinflammatory and prothrombic states. Moreover, endothelial dysfunction contributes to atherosclerotic plaque initiation and progression, leading to atherosclerosis [12]. Stress, hypertension, infection, hyperlipidemia, and hyperglycemia would lead to endothelial dysfunction [11,13]; the generation of ROS and the antioxidant defense system is the main reason for endothelial dysfunction [11].

Palmitic acid (PA) is a long-chain saturated fatty acid that is the most common saturated fatty acid found in modern diets [14]. A Canadian cohort study collected plasma samples from young, healthy adults and demonstrated that palmitic acid is the major FA, ranging from 0.3 to 4.1 mmol/L in plasma concentration. In addition, regardless of gender or race, PA is the most abundant fatty acid in plasma samples [15]. In clinical settings, Kawasaki disease (KD) patients with coronary artery lesions (CALs) showed significantly higher PA in differentially expressed metabolites compared to non-coronary artery lesion patients. PA can potentially be a diagnostic biomarker in KD patients with CALs [16]. In addition, PA increases IL-1β, CCL3, CCL4, SPP1, IRF5, MMP7, and MMP9 mRNA levels, which are vital genes that correlate with late-stage carotid atherosclerosis [17]. Thereby, PA has been used in most studies to investigate fatty acid-induced vascular lipotoxicity.

Lipotoxicity is the accumulation of excess fatty acids in parenchymal cells of various tissues, leading to disruption of cellular homeostasis and cellular dysfunction [18]. These ectopic fat deposits may act as an active mechanism to release multiple biochemical mediators in the body that affect insulin resistance and inflammation, both of which increase cardiovascular risk [19]. Lipotoxicity affects various organs, including cardiac and vascular tissue. In the U.S., approximately 25% of adults were affected by lipotoxicity through obesity, diabetes, and heart failure [20]. Accumulated evidence demonstrated that circulating free fatty acids like palmitic acid (PA) can induce endothelial dysfunction through disruption of insulin signaling, promoting oxidative stress and inflammation, impairing mitochondrial integrity, and inducing ER stress. These different mechanisms, such as PA-promoted acid sphingomyelinase (ASM), which increase superoxide production via NOX2 and NOX4, suppress the IRS/PI3k/Akt/eNOS signaling pathway, leading to insulin resistance of vascular endothelium in in vivo and in vitro studies [21]. PA-induced fragmentation of mitochondria leads to malfunction of mitochondria and loss of mitochondrial membrane potential through inhibition of the tyrosyl transfer-RNA synthetase/poly (ADP-ribose) polymerase 1 (TysR/PARP1) pathway, which reduced expression of MFN1, MFN2, and OPA1, leading to increased ROS [22]. PA induced ER stress and activated unfolded protein response (UPR); meanwhile, ER stress causes endothelial mesenchymal transition (EndMT), inhibits angiogenesis (cell migration, tube number), and decreases cell viability [23].

This review not only revealed molecular signaling such as PA but could induce apoptosis through suppressed κ-opioid receptor (κ-OR). In addition, PA induced inflammation by inhibiting the expression of Dickkopf-1 (DDK1). PA also induces autophagy but disturbs autophagy flux by inhibiting the fusion of autophagosome and lysosome. In ROS, PA inhibited liver kinase B1 (LKB1)-dependent AMPK/Nrf-2 signaling pathway to increase intracellular ROS content. PA inhibits sarco/endoplasmic reticulum calcium ATPase (SERCA) expression and Ca^2+^ pump ATPase activity, which induces ER stress and activates JNK. The molecular mechanisms from recent evidence are summarized in Figure 1.

Also, this review disclosed various therapeutic agents against vascular endothelial lipotoxicity through different targeting, such as traditional Chinese medicine. Salvianolic acid B (SAB), natural polyphenol honokiol, natural flavonoid dihydromyricetin (DHM), and isopropyl 3-(3,4-dihydroxy phenyl)-2-hydroxy propionate (IDHP) were used to attenuate PA-induced vascular apoptosis, pyroptosis, and ferroptosis, respectively. Unsaturated fatty acids like eicosapentaenoic acid (EPA) and docosahexaenoic acid (DHA) attenuated inflammation through reduced inflammation cytokines. Resveratrol (RSV), a natural phenol, was used to restore autophagy and improve autophagy flux disturbed by PA. Protocatechuic acid (PCA, 3,4-dihydroxybenzoic acid) is the major metabolite of polyphenols activating the LKB1/AMPK/Nrf-2 pathway to reduce ROS generation. DAPA increases mitochondrial viability, ATP production, mitochondrial biogenesis, and reduces apoptosis of PA-treated endothelial cells through the activated SIRT1/PGC-1α signaling pathway. Ilexgenin A suppresses PA-induced ER stress by activating AMPK to inhibit TNXIP/NLRP3-related IL-1b secretion. Cyanidin-3-O-glucoside (C3G) could regulate phosphorylation of p-IRS and subsequently activate the Akt/eNOS pathway to restore NO production through inhibition of IKK, and JNK is Nrf2 dependent.

This review describes the molecular mechanism of vascular lipotoxicity caused by PA and the agents that can be used for prevention or treatment. It also provides researchers with future research directions and clinical applications in cardiovascular disease.

## 2. The Molecular Mechanisms and Preventive Strategies of Fatty Acids Induced Vascular Endothelial Dysfunction

Free fatty acids modulate endothelial dysfunction through programmed cell death, inflammation, autophagy, mitochondrial dysfunction, ROS, ER stress, and insulin resistance in endothelial cells. However, several preventive approaches have been investigated by disclosing molecular mechanisms to counteract fatty acid-induced vascular dysfunction.

### 2.1. Apoptosis/Necroptosis/Pyroptosis/Ferroptosis

Palmitic acid (PA) induced apoptosis of endothelial cells by increasing caspase-3 and Bax and decreasing Bcl-2 through suppressed κ-opioid receptor (κ-OR) regulated PI3K/AKT/eNOS signaling pathway [16]. Moreover, PA induced vascular endothelial cell apoptosis by activated TNF receptor type 1 (TNF-R1)/caspase-8 signaling pathway [17]. PA also induced apoptosis of endothelial cells through increased expression of serpin family E member 1 (SERPINE1), caspase-3, and inhibition of signal transducer and activator of transcription 3 (STAT3) [18]. PA could induce p53-regulated caspase-3-mediated apoptosis of endothelial cells through inhibiting Sirt-1 expression [19]. Moreover, PA-enhanced expression of pentraxin 3 (PTX3) regulated apoptosis by activating the IKK/IKb/NF-kB signaling pathway [20].

Gan et al. found that PA inhibited the expression of Dickkopf-1 (DDK1), which regulated cysteine-rich angiogenic inducer 61 (CCN1) and Wnt/B-catenin, leading to inflammation and apoptosis of HUVEC [24]. Chen et al. demonstrated that ATF 4 deficiency aggravated high-fat-diet-decreased limb perfusion, angiogenesis, and wound closure in an in vivo study. In addition, PA inhibited expression of ATF4, leading to hypomethylation of H3K4 on NOS3 and ERK1. Overexpression of ATF4 decreased PA-induced apoptosis, promoted cell proliferation and tube formation, and also ATF4 maintained methylation of NOS3 and ERK1 through formation of a complex with methionine adenosylmethyltransferase 2A (MAT2A) and lysine methyltransferase 2A (MLL1) [25].

Wang et al., using proteomic scale profiling, discovered two transcription factors, v-maf musculoaponeurotic fibrosarcoma oncogene family protein G (MAFG) and v-maf musculoaponeurotic fibrosarcoma oncogene family protein F (MAFF), which regulated PA-induced early apoptosis [26]. PA also suppressed AMPK, leading to the induction of lipoapoptosis, ER stress, and increased cellular adhesion molecular (ICAM, E-selectin) expression [27].

PA was reported to have induced receptor-interacting protein kinase-3 (RIPK3) and carboxyl-terminal hydrolase (CYLD)-dependent programmed necrosis of endothelial cells through activated Ca^2+^-regulated autophagy and depletion of ATP [28]. Hu et al. found that PA induced pyroptosis of endothelial cells through activating the NLRP3 inflammasome/caspase-1 signaling pathway by increasing mitochondrial ROS [29]. In addition, Martino et al. discovered that PA suppressed SIRT3 to activate NLRP3/caspase-1 signaling pathway to induce endothelial cell pyroptosis [30].

PA also triggered ferroptosis of endothelial cells through promoted intracellular Fe^2+^ accumulation and inhibited the cyst(e)ine/GSH/GPX4 axis by suppressing SLC7A11, GSH, and GPX4 expression [31].

#### Therapeutic Strategies: Targeting to Attenuate the Programmed Cell Death

Salvianolic acid B (SAB) is derived from *Salvia miltiorrhiza* Bunge, which belongs to traditional Chinese medicine. SAB has neuroprotective properties and is also a ROS scavenger. Zhai et al. reported that cotreatment with salvianolic acid B (SAB) attenuated p53-related apoptosis of endothelial cells by enhanced expression of Sirt-1 and BCL-2 expression; meanwhile, SAB reduced ROS generation and TXNIP expression and reversed mitochondrial membrane potential [32]. This result showed that SAB may be a potential drug to treat cardiovascular disease.

Honokiol (HNK) is a natural polyphenol derived from *Magnolia officinalis*. A recent study found that HNK inhibited histone deacetylase (HDAC6) to maintain cystathionine γ-lyase (CSE) protein stability, which subsequently ameliorates angiotensin II (Ang II)-induced endothelial dysfunction and hypertension [33]. Honokiol attenuated apoptosis and restored NO generation of endothelial cells by inhibiting the NK-kB pathway-related PTX3. Moreover, honokiol demonstrated anti-inflammatory ability by decreasing IL-6, IL-8, and MCP-1 expression [34]. Therefore, PTX3 can be a therapeutic target for reducing atherosclerosis in clinical applications.

Dihydromyricetin (DHM) is a natural flavonoid derived from *Ampelopsis grossedentata*. Its antioxidant, anti-inflammation, and antitumor effects were reported. Pretreatment with dihydromyricetin (DHM) attenuated pyroptosis induced by PA through activation of the Nrf2 pathway, reducing intracellular ROS and mitochondrial ROS, leading to decreased lactate dehydrogenase (LDH) and IL-1B levels [29]. These findings provide a new avenue as DHM may prevent atherosclerosis induced by lipotoxicity.

Clopidogrel is a thienopyridine adenosine diphosphate (ADP) receptor inhibitor. It is widely used to treat cardiovascular patients and to prevent ischemic cerebrovascular disease and peripheral vascular disease. Wang et al. discovered that silencing of long non-coding RNA HIF1A-AS1 (HIF1 alpha-antisense RNA 1) decreased PA-induced apoptosis by characterized reduced expression of caspase-3 and caspase-9 and increased BCL-2 expression. Clopidogrel could enhance cellular viability and decrease apoptosis by inhibiting HIF1A-AS1 expression [35]. These results showed HIF1A-AS1 may be a novel molecular mechanism to fight against cardiovascular disease induced by hyperlipidemia.

In contrast to saturated fatty acids, polyunsaturated fatty acids could reduce coronary heart disease [36]. Docosahexaenoic acid (DHA) is an ω-3 long-chain polyunsaturated fatty acid that is highly present in some fish like herring, trout, and salmon. Cotreatment with docosahexaenoic acid (DHA) enhanced miR-3691-5p to suppress serpine1 and caspase-3 to attenuate apoptosis of endothelial cells [37]. In addition, oleic acid, as a monounsaturated fatty acid, attenuated saturated fatty acid-induced inflammation in human aortic endothelial cells by suppressing NF-kB activation from a previous study [38]. Lee et al. showed that cotreatment with monounsaturated fatty acids like oleate (Ole) and palmitoleate (PO) attenuated lipoapoptosis and cellular adhesion molecular through the AMPK pathway in endothelial cells. Several steps of this nature study were removed as a limitation of these findings; thus, the implications of these findings on clinical inferring should be prudent [27].

Isopropyl 3-(3,4-dihydroxyphenyl)-2-hydroxypropionate (IDHP) is a metabolite of the traditional Chinese medicine Dansheng. A recent study showed that IDHP has anti-inflammatory and cardioprotective effects [39]. He et al. demonstrated that IDHP can reduce intracellular ferrous accumulation, ROS production, and lipid peroxidation through reverse GPX4 and xCT/SLC7A11 expression to attenuate ferroptosis induced by PA. IDHP also can suppress cell cycle and cellular senescence under lipotoxicity [40]. This result provided a therapeutic idea with IDHP against vascular lipotoxicity-induced senescence.

The mechanisms targeted in programmed cell death are summarized in Figure 2.

### 2.2. Inflammation

PA-augmented expression of long-chain acyl-CoA synthetase (ACSL1, ACSL3), which subsequently activates the NF-kB pathway and P21 that culminate in endothelial dysfunction through increasing adhesion molecules, inflammatory cytokines, and inflammatory proteins [41]. PA induced intracellular ROS generation through NOX2 and NOX4, which activated the IKKb/NF-kB pathway to increase TNF-α, IL-6, and ICAM-1 expression, leading to endothelial cell inflammation [42]. Bharat et al. also found that PA-enhanced monocyte bindings, adhesion molecular expression (ICAM1, VCAM1, E-selectin), and inflammatory chemokine (IL-8, MCP1) induce endothelial inflammation through NOX4-mediated ROS production [43].

Rao et al. discovered increased nucleophosmin (NPM) levels in human carotid atherosclerotic plaques, mostly in endothelial cells. Furthermore, NPM activated NF-kB and promoted NF-kB downstream genes (IL-6, ICAM-1, and E-selectin), leading to inflammation of HUVEC. Knockdown of NPM could inhibit atherosclerotic plaque formation [44].

PA-activated toll-like receptor 4 (TLR4) and its downstream signaling pathway of MyD88/NF-kB, TRIF/AMPK/JNK, p-38, ERK 1/2, and TRAF 6 lead to inflammation [45]. PA also promoted inflammatory mediators MCP-1 and ICAM-1 by activating the TLR4-dependent MyD88 and TRIF pathways. Meanwhile, PA upregulated the inflammasome NLRP3/caspase-1 signaling pathway to increase expression of IL-1B, which ultimately induced vascular inflammation [46].

Canonical toll-like receptors (TLRs) like lipopolysaccharide (LPS), Pam3-Cys-Ser-Lys4 (Pam3CSK4), or macrophage-activating lipopeptide-2 (MALP2) induced a proinflammatory response at early treatment (20 min, 30 min, 60 min, 120 min). Different from other canonical TLRs, PA promoted a proinflammatory response by increasing the expression of IL-1B, NF-kB, p-JNK, and monocyte adhesion through acyl-CoA synthetase-1 (ACSL1) after exposure at 6 h in HAEC cells. This inflammatory response suppressed insulin-regulated vasodilation but not nitroprusside-related vasodilation in an ex vivo study. This study suggested PA-induced inflammation, leading to vascular dysfunction distinct from the canonical TLR pathway. Regulation of COX2 by other ACSL isotypes could be conducted to clarify the inflammation caused by PA in endothelial [47].

PA induced endothelial inflammation through increasing ICAM1 expression and monocyte adhesion by activating the cGAS/STING/IRF3 signaling pathway. Meanwhile, the high-fat-diet-fed STING deficiency mice reduced macrophage infiltration in adipose tissue and vascular ICAM-1 expression. These findings demonstrated that the mtDNA-cGAS-STING-IRF3 signaling pathway is related to metabolic stress-induced endothelial dysfunction, which could be a potential candidate against vascular lipotoxicity [48].

#### Therapeutic Strategies: Targeting to Attenuate the Inflammation

Eicosapentaenoic acid (EPA) is a long-chain unsaturated fatty acid present in fish oil; it effectively reduces coronary events in Japanese hypercholesterol patients [49]. Ishida revealed cotreatment with EPA ameliorate endothelial dysfunction via inhibiting expression of long-chain acyl-CoA synthetase (ACSL) by following suppressed NK-kB signaling pathway and p21 to decrease inflammatory-related cytokine, adhesion molecular expression [41]. These findings revealed the critical role of unsaturated fatty acid EPA in an anti-atherogenic diet related to cardiovascular disease.

Docosahexaenoic acid (DHA) was reported to reduce cardiovascular risk by inhibiting endothelial cell inflammation and preventing atherosclerosis formation [50]. Abriz et al. showed that DHA reduced accumulation of palmitic acid, increased cell survival rate, decreased NO production and inflammatory cytokines (TNF-α, IL-6, and NF-κB), and decreased apoptosis after PA treatment in an in vitro study [51]. Like EPA, DHA has an anti-atherosclerotic change in endothelial cells that prevents cardiovascular disease.

Adiponectin (APN) is an adipocyte-secreted hormone that reduces ROS generation through suppressed expression of NOX2 and follows the NF-kB pathway to diminish inflammatory cytokines (TNF-α, IL-6, ICAM-1). Meanwhile, pretreatment with APN restored the IRS-1/AKT/eNOS/NO signaling pathway to ameliorate PA-induced vascular insulin resistance [42]. These results disclosed that APN has a protective effect on cardiovascular disease by improving endothelial dysfunction.

Pretreatment with blueberry metabolite mitigated endothelial inflammation induced by PA via inhibition of NOX4 and increasing NO production. This result provided a concept that blueberries might be a food complement to lessen lipotoxicity-induced vascular dysfunction [43].

Clinopodium Chinense (CCE) has anti-inflammatory, antitumor, antioxidant, anti-cardiotoxicity, and antiradiation properties even though it was used to treat diabetes by the Fujian people of China. A previous study showed that an extract of CCE reduced the high glucose-induced apoptosis of HUVEC by regulating the pAkt/NF-κB/Bax/Bcl-2/caspase-3 pathway [52]. Shi et al. revealed that pretreatment with ethyl acetate CCE inhibited toll-like receptor 4 (TLR4), NF-kB, and AMPK expression, eliminated inflammation, and restored tyrosine phosphorylation and insulin-mediated vasodilation signaling pathways in in vivo and in vitro studies. This result showed that CCE can potentially treat inflammation and insulin resistance induced by lipotoxicity in vascular endothelial [45].

Homoplantaginin is a flavonoid derived from Salvia plebeia R. Br., which belongs to traditional Chinese medicine. Previous studies showed that homoplantaginin suppressed PA-induced inflammation and insulin resistance in the endothelial cells by modulating the IKKβ/IRS-1/pAkt/peNOS pathway [53]. Moreover, pretreatment with homoplantaginin attenuated PA-induced vascular inflammation by suppressing the expression of inflammatory cytokines (MCP-1, ICAM-1, IL-1B) through inhibiting TLR4 and NLRP3 signaling pathways. Moreover, homoplantaginin reversed PA-disturbed NO generation [46]. Therefore, homoplantaginin can be a candidate for treating or preventing vascular disease.

Apigenin is a natural flavone that is abundantly found in many vegetables and fruits. Apigenin has antioxidant, anti-inflammatory, and anti-cancer effects [54]. Miao et al. demonstrated that apigenin inhibits proinflammatory cytokines (NO, TNF-α, IL-6), promotes NO production and mitochondrial membrane potential (MMP), attenuates cell adhesion molecules (ICAM, VCAM) and cell death, and enhances the expression of FOXO1 after PA treatment in human aortic endothelial cells (HAEC) [4]. These data demonstrated that apigenin can be used against vascular lipotoxicity as an alternative nutritional support.

Withaferin A (WA) is the active component of Withania somnifera; it is a steroidal lactone and NF-kB inhibitor through hyperphosphorylation of IKKβ [55]. Batumalaie et al. demonstrated that pretreatment with withaferin A (WA) abolished PA-induced increases in inflammatory cytokines (TNF-α, IL-6) via inhibition of the IKKB/NF-kB pathway. Meanwhile, WA restored NO production by altering the phosphorylation of IRS, consequently affecting the AKT/eNOS pathway. WA also suppresses ROS generation and endothelin-1 (ET-1) and plasminogen activator inhibitor type-1 (PAI-1) expression [56]. These results demonstrated that WA has antioxidant and anti-inflammation effects against lipotoxicity-induced endothelial dysfunction.

Myeloid cell-derived growth factor (MYDGF) decreased atherosclerotic plaque formation and ameliorated endothelial apoptosis, inflammation, and adhesion by inhibiting the PKC/MAP4K4/NF-kB signaling pathway in in vivo and in vitro studies [57]. MYDGF can serve as a therapeutic target for treating metabolic disorders and atherosclerosis.

The mechanisms targeted in inflammation are summarized in Figure 3.

### 2.3. Autophagy

PA activated autophagy to promote ROS production, leading to inhibition of NO production, cell migration, and tube formation through the calcium ion/protein kinase Cα/nicotinamide adenine dinucleotide phosphate oxidase 4 (Ca^2+^/PKCα/NOX4) pathway in the HUVEC [58]. PA could activate autophagy but suppress JNK, p38-dependent autophagic flux, which leads to cell apoptosis [59]. Similarly, Lee et al. found PA could induce autophagy but disturbed autophagy flux by inhibiting the fusion of autophagosome and lysosome [60]. PA also caused oxidative stress on endothelial cells by suppressing the expression of transcription factor EB (TFEB), which subsequently disturbed autophagy flux by inhibiting lysosomes [61]. In contrast, Song et al. discovered that PA inhibits AMPK/mTOR-regulated autophagy, which subsequently promotes intracellular ROS production and decreases the expression of p-eNOS [62]. These results suggested that PA-regulated autophagy plays a two-pronged influence on endothelial function.

PA promoted ANGPTL4 expression, and overexpression of ANGPTL4 attenuated PA-impaired cell viability; meanwhile, ANGPTL4 regulates cellular proliferation through modulating autophagy [63].

#### Therapeutic Strategies Targeting on Autophagy

Resveratrol (RSV) is a natural phenol derived from plants like grapes, blueberries, and raspberries. It has antioxidant and anti-inflammatory effects [64]. Song et al. used RSV to restore autophagy by activating AMPK to attenuate ROS production and enhance p-eNOS, which improved endothelial dysfunction caused by PA [62]. Moreover, pretreatment with RSV improved autophagy flux via enhanced transcription factor EB (TFEB) to attenuate oxidative stress and endothelial injury [61]. These findings implicated the protective role of resveratrol in treating or preventing atherosclerosis in clinical application.

PA induced senescence of endothelial cells by evidence of increased senescence-associated acidic β-galactosidase (SA-β-GAL) and reduced hyperphosphorylated retinoblastoma gene product (ppRb). C1q/tumor necrosis factor-related protein 9 (CTRP9) promoted autophagy through AMPK and improved autophagy flux, ameliorating endothelial senescence induced by PA in HUVEC [60]. These results demonstrated that CTRP9 exerts an anti-atherosclerosis effect by reducing cellular senescence. An in vivo study can be conducted to ascertain CTRP9’s role in treating atherosclerosis.

### 2.4. Reactive Oxygen Species (ROS)

PA enhanced ROS production by promoting NADPH oxidase activity and increasing the expression of NOX2 and NOX4 [65]. Li et al. revealed that PA promoted acid sphingomyelinase (ASM), which increases superoxide production via NOX2 and NOX4 to suppress the IRS/PI3k/Akt/eNOS signaling pathway, leading to insulin resistance of vascular endothelial in in vivo and in vitro studies [21]. PA-disturbed potassium channel (K_Ca_2.3) regulated endothelium-dependent hyperpolarization (EDH)-type relaxation in rats’ superior mesentery artery. In addition, PA inhibits the expression of K_Ca_2.3 and decreases K_Ca_2.3 current density through the ROS-related NADPH oxidase (NOX)/p-38/NF-kB signaling pathway in HUVEC [66]. These findings revealed that inhibition of NOX can potentially ameliorate PA-impaired vascular relaxation resistance, and ASM can be a clinical biomarker and therapeutic target of diabetic vascular.

PA inhibited liver kinase B1 (LKB1)-dependent AMPK/Nrf-2 signaling pathway to increase intracellular ROS content [67]. Han et al. similarly found that PA suppresses expression of LKB1 and its downstream AMPK/Nrf-2/PGC1-α signaling pathway to induce oxidative stress and disturb mitochondrial biosynthesis [68]. PA could impair expression of glucagon-like-peptide-1 receptor (GLP-1R) and protein kinase A (PKA), leading to increased ROS production and reduced NO and p-eNOS in in vivo and in vitro studies [69].

PA induced ROS generation and vascular fibrosis through myeloid differentiation 2 (MD2) by suppressing the AMPK/Nrf-2 pathway. MD2 knockout mice fed with a high-fat diet (HFD) showed decreased vascular oxidative stress and vascular fibrosis by evidence of reduced collagen I, collagen III, α-SMA, and TGF-β expression; meanwhile, silencing of MD2 or pretreatment with anti-MD2 promoted antioxidant enzyme and expression of AMPK and inhibited fibrosis markers in rat aortic endothelial cells [70].

Recently, Zhu et al. disclosed that PA promoted intracellular ROS accumulation, which aggravated endothelial senescence by evidence of acetylcysteine (NAC) lessening SA-β-gal activity, pRB, and p16 levels [16].

Zhang et al. found that PA suppressed the AMPK/KLF2/eNOS signaling pathway. Overexpression of KLF2 or enhanced AMPK with AICAR could restore AMPK/KLF2/eNOS signaling, meanwhile reducing ROS production of cardiac microvascular endothelial cells (cMECs). In addition, promoting AMPK or KLF2 in an in vivo study could increase coronary flow and decrease leukocyte adhesion [71].

PA upregulated monoamine oxidase B (MAOB) to induce ROS generation and inflammation, increase cellular adhesion cytokines, and reduce apoptosis of endothelial cells. Moreover, Tian et al. discovered that miR-3620-5p could regulate MAOB expression. Using MAOB inhibitors, selegiline, or knockdown of MAOB with siRNA suppressed ROS production, reduced proinflammatory cytokines, and reduced apoptosis in in vitro studies. Meanwhile, selegiline attenuated atherosclerotic lesions and modulated the composition and function of the gut microbiome in an in vivo study. This result declared that inhibiting MAOB, like selegiline, could be promising in preventing atherosclerotic cardiovascular diseases [72].

#### Therapeutic Strategies Targeting on Reducing ROS

Heparan sulfate is a glycosaminoglycan that contributes to angiogenesis, wound healing, and inflammation processing [73]. The small molecule glycomimetics (C1–C4) are designed to mimic glycosaminoglycan biological function. Mahmoud et al. discovered that pretreatment or cotreatment with glycomimics (C1~C4) inhibited NADPH oxidase activity, which reduced ROS generation. Moreover, glycomimics promoted Nrf2-dependent antioxidant enzymes (NQO1, HO-1, SOD, CAT) and restored Akt/eNOS-related NO production to diminish oxidative damage and endothelial dysfunction in in vitro and in ex vivo studies [65]. These findings implicated that small molecule glycomimics could prevent or treat oxidative stress-related disease.

Phloretin is a dihydrochalcone that belongs to the natural phenol class. It has antioxidant, anti-inflammation, and anti-cancer properties. Pretreatment with phloretin diminished PA-induced oxidative stress through the LKB1/AMPK/Nrf-2 pathway by promoting antioxidant enzyme SOD and GPx expression and restoring mitochondrial membrane potential simultaneously in endothelial cells [67]. PA enhanced mitochondrial ROS (mtROS) production by increasing the acetylation of Mn-superoxide dismutase (MnSOD) through inhibition of AMPK/Sirt3. Cotreatment with phloretin promoted the expression of AMPK/Sirt3 to decrease MnSOD’s acetylation to inhibit mtROS after PA exposure in in vivo and in vitro studies. PA increases lysine acetylation, which inhibits MnSOD; phloretin promotes the expression of Sirtuin 3 (Sirt3), which is a lysine deacetylase that restores the antioxidant properties of endothelial cells [74]. These results provided a novel antioxidant activity of phloretin through the AMPK pathway and showed it to be a food additive against vascular lipotoxicity. Further, Li et al. discovered that phloretin could enhance total antioxidant capacity, glutathione reductase activity, and mitochondrial membrane potential through inhibition of LncBAG6-AS to ameliorate oxidative stress induced by PA in endothelial cells [75].

Protocatechuic acid (PCA, 3,4-dihydroxybenzoic acid) is the major metabolite of polyphenols abundant in fruits and vegetables [76]. Pretreatment with PCA mitigated oxidative stress by reducing ROS generation through activating the LKB1/AMPK/Nrf-2 pathway and reversed AMPK-dependent PGC1-α to enhance mitochondrial biosynthesis against lipotoxicity-induced oxidative stress in endothelial cells [68]. These results demonstrated PCA as a promising therapeutic candidate for vascular lipotoxicity; however, an in vivo study could be conducted to ascertain its protective role.

Ke et al. reported that metformin pretreatment activated the AMPK signaling pathway to reduce ROS production and restore NO level and p-eNOS protein expression by enhancing GLP-1R and PKA in PA- or HFD-induced vascular dysfunction. Pretreatment with metformin and liraglutide also protects the endothelial cell from lipotoxicity through GLP-1R and PKA signaling pathways [69].

Liraglutide is a human glucagon-like peptide-1 (GLP-1) analog; it can fight against the rapid degradation of dipeptidyl peptidase 4. Liraglutide reduced the incidence of cardiovascular disease (CVD) and CVD-related fatal rates from previous studies [77]. Cotreatment with liraglutide recovered PA-impaired endothelial tubes by enhancing the PI3K/Akt/Foxo-1/GTPCH1 signaling pathway and NO production [78]. PA promoted ROS production and apoptosis and inhibited the expression of ET-1 in immortal mouse islet microvascular endothelial cells (MS-1), liraglutide inhibited ROS production and cell apoptosis, and enhanced the expression of ET-1 through the GLP-1R/PKA and GTPCH1/eNOS signaling pathways [79]. Liraglutide, therefore, has an antioxidant effect by ameliorating oxidative stress.

Hot water extract of Loliolus beka gray meat (LBM) is rich in taurine. Pretreatment with LBM could ameliorate PA-induced proinflammatory cytokine (IL-1B, TNF-α, COX-2) and overproduction of ROS in HUVEC. Therefore, the extract of LBM could be supplemented to protect vascular endothelial from lipotoxicity [80].

Danzhi jiangtang capsule (DJC) is a traditional Chinese medicine compound used to treat diabetes for years. Lu et al. showed DJC attenuated ER stress and oxidative stress, diminished cell apoptosis, restored aortic relaxation by increasing p-eNOS expression, promoted antioxidant properties, and decreased inflammatory cytokine during vascular lipotoxicity in in vivo and in vitro studies. These findings illuminate that DJC has endothelial protection against endothelial dysfunction under lipotoxicity [81].

Vanillic acid (VA) is a phenolic metabolite of anthocyanins; it could reduce hyperinsulinemia, hyperglycemia, and hyperlipidemia and attenuate inflammation in high-fat diet-fed rats from previous reports [69]. Ma et al. disclosed that VA reduced the production of ROS by activating the LKB1/AMPK/Nrf-2 pathway, which promotes antioxidant enzyme (CAT, SOD) activities and reverses mitochondrial membrane potential by regulating SIRT-1/PGC1-α during PA-induced vascular lipotoxicity. An in vivo study could be conducted to validate pharmacokinetics and other functions of VA in the future [82].

Geraniol (GOH) is a monoterpenoid alcohol that is a significant ingredient of palmarosa oil, lavender, and rose oil. GOH has been reported as GOH-activated Nrf-2 to diminish oxidative stress induced by an atherogenic diet in an in vitro study [71]. Wang et al. said that GOH improved acetylcholine-dependent vascular relaxation, downregulated NOX2-derived ROS generation, and suppressed adhesion molecules (ICAM-1, VCAM-1) to attenuate vascular inflammation in HFD-fed mice and PA-treated endothelial cells [83]. GOH, therefore, has an antioxidant effect, and future studies could specifically examine its impact on vascular tone.

Delta-valerobetaine (δVB) is a constitutive metabolite of ruminant milk and is produced in the rumen from the free dietary Nε-trimethyllysine common in the plant kingdom. It has antioxidant, anti-inflammatory, and anti-cancer properties from previous studies. Pretreatment with δVB diminished mitochondrial ROS production, pyroptosis, and cytotoxicity of endothelial cells; meanwhile, δVB promoted SIRT2, SIRT3, SIRT4, and SIRT5 expression and attenuated insulin resistance [30]. These results indicated that δVB has an antioxidant effect, and SIRT3 could be developed as a novel antioxidant approach.

Canagliflozin (CAN) is a sodium-dependent glucose transporter 2 (SGLT2) inhibitor that is used to treat type 2 diabetes, which increases urinary glucose excretion. CAN has cardiovascular protective effects on palmitic acid-induced cardiomyocytes from previous studies. Hao et al. disclosed that CAN suppressed ROS-dependent P38/JNK signaling pathway to decrease senescence-associated secretory phenotype (SASP), DNA damage, and the cell cycle of endothelial cells [84]. These results show that CAN is a novel therapeutic medication for patients with hyperlipidemia.

Meteorin-like protein (Metrnl) is a novel adipokine produced in adipose tissue and skeletal muscle. It has anti-inflammatory effects and reduces the production of reactive oxygen species from previous studies. Furthermore, Metrnl levels were inversely associated with CAD risk, fatty acid levels, and inflammatory cytokine expression. Liu et al. discovered that Metrnl was found to reduce PA-induced ROS production, which in turn inhibited NLRP3 inflammasome expression and the inflammatory cytokine-like IL-1B, thereby improving endothelial aerobic aspiration [85]. Metrnl, therefore, has a therapeutic role in defending CAD patients against endothelial dysfunction.

NG-R1 is the most potent notoginsenoside (NGs) and is the main bioactive compound of Panax notoginseng. It has been shown to have cardiovascular, hepatoprotective, and neuroprotective activities as well as antidiabetic and anti-cancer activities [86]. Liu et al. found that NG-R1 could reduce ROS production; meanwhile, it could restore glucose uptake impaired by PA through upregulating Nrf2 to suppress phosphorylation of IRS-1 and elevated phosphorylation of Akt, which could alleviate insulin resistance. NG-R1 also promoted Nrf2-related antioxidant enzymes like NAD(P)H:quinone oxidoreductase 1 (NQO1), heme oxygenase-1 (HO-1), and glutathione peroxidase (GPx) to maintain redox balance in endothelial cells [87]. This study revealed that NG-R1 has the potential to be an additional food to mitigate vascular disease and metabolic disorders.

Physcion is an anthraquinone extracted from *Rheum palmatum* L. Previous studies have shown that it can decrease very low-density lipoproteins and has antioxidant, anti-inflammatory, anti-bacterial, anti-viral, and anti-cancer properties. Wang et al. disclosed that physcion reduced cellular apoptosis, ROS production, inflammation, and ER stress and reversed NO, p-eNOS, and Nrf2 expression, which were impaired by PA. Meanwhile, physcion elevated vasodilation and reduced plasma triglyceride (TG) in an in vivo study. These implicated physcion as a possible agent to prevent cardiovascular disease and endothelial dysfunction [88].

The mechanisms targeted in ROS are summarized in Figure 4.

### 2.5. Mitochondrial Dysfunction

Broniarek et al. showed that chronic exposure to high PA (100/150 uM) in the EA hy926 human endothelial cell line impaired mitochondrial function through increasing intracellular and mitochondrial ROS, altering mitochondrial membrane potential, reducing mitochondrial respiration, decreasing ATP synthesis, and increasing fatty oxidation. All of these results caused endothelial dysfunction and reduced cell viability [89].

PA reduces mitochondrial membrane potential and induces mitochondrial damage, releasing mitochondrial DNA to the cytosol, which is converted to cGAMP by cyclic GMP-AMP synthase (CGAS). cGAMP subsequently activates the STING/IRF3/MST1 signaling pathway, which enhances the phosphorylation of Yes-associated protein (YAP), suppressing the angiogenesis of endothelial cells. Mitochondrial damage could induce endothelial dysfunction, which implies that protection of the mitochondria is crucial in maintaining the angiogenesis capacity of endothelial cells [90].

PA activated the cGAS-STING pathway by mitochondrial DNA that increased cell migratory capacity and ultimately induced endothelial-to-mesenchymal transformation (EndMT), which involved oxidative stress and inflammation of endothelial cells [91].

PA induced mitochondrial fission by increased expression of Drp1 and decreased Mfn2. Meanwhile, PA promoted the loss of mitochondrial membrane potential [92]. Yang et al. further showed that PA-induced fragmentation of mitochondria leads to malfunction of mitochondria and loss of mitochondrial membrane potential through inhibition of the tyrosyl transfer-RNA synthetase/poly (ADP-ribose) polymerase 1 (TysR/PARP1) pathway, which reduced expression of MFN1, MFN2, and OPA1, leading to increased ROS [22].

An et al. discovered that PA promoted IQ motif-containing GTPase-activating protein (IQGAP1) expression, thereby increasing mitochondrial ROS accumulation and releasing mitochondrial DNA (mtDNA) into the cytoplasm. Subsequently, cGAS-STING was activated to promote inflammation and pyroptosis through an increase in the expression of caspase-1, GSDMD, IL-1B, and IL-18 in in vitro and in vivo studies [93].

#### Therapeutic Strategies Targeting on Maintaining Mitochondrial Function

D-chiro inositol (DCI) was reported to have reduced endothelial dysfunction and ROS production in hyperglycemic rats [94]. DCI-enriched Tartary buckwheat bran extract (TBBE) suppresses oxidative stress by lowering superoxide anion generation. Moreover, TBBE protects mitochondrial morphology and integrity by regulating phosphorylation of dynamin-related protein 1 (Drp1), enhances mitochondrial membrane potential, and prevents the release of cytochrome C. TBBE could inhibit ER stress and JNK-associated inflammation and diminish apoptosis by decreasing caspase-3 in vascular endothelial cells [92]. This study indicated that TBBE could be developed to counteract endothelial inflammation and vascular disease.

Moreover, cotreatment with resveratrol (RSV) activated the TysR/PARP signaling pathway to suppress ROS generation, which reversed the expression of MFN1, MFN2, and OPA1 sequentially to maintain mitochondrial morphology and mitochondrial membrane potential to preserve endothelial function during lipotoxicity. These outcomes opened a new view to further study for atherosclerosis induced by fatty acid based on the TysR/PARP1 pathway [22].

Dapagliflozin (DAPA) is a sodium-dependent glucose transporter 2 (SGLT-2) inhibitor with protective effects on the heart, kidney, and liver. He et al. discovered that cotreatment with DAPA increases mitochondrial viability, ATP production, mitochondrial biogenesis, and reduces apoptosis of PA-treated endothelial cells through the activated SIRT1/PGC-1α signaling pathway [95]. Therefore, DAPA has the potential as a novel medication for treating hyperlipidemia.

The transient receptor potential (TRP) melastatin-like subfamily member 4 (TRPM4) is a non-selective cation channel that mediates the transport of Na+ and K+ across the plasma membrane into the cytoplasm. TRPM4 is also an important Ca^2+^ regulator, which may be involved in the cardiac conduction system and related to the occurrence of arrhythmia [96]. Xue et al. showed that PA enhanced TRPM4 expression in endothelial cells, and following silencing of TRPM4, apoptosis, inflammatory cytokine expression, intracellular calcium accumulation, mitochondrial membrane potential, and ROS level were reduced. miR-133a-3p could bind to the TRPM4 3’UTR region, which directly suppressed TRPM4 expression to ameliorate mitochondrial dysfunction and ROS production-related endothelial injury induced by PA. These results showed that miR-133a-3p could be developed as a potential target in preventing endothelial injury [97].

The mechanisms targeted in mitochondrial are summarized in Figure 5.

### 2.6. ER Stress

PA induced ROS-associated endoplasmic reticulum (ER) stress, as evidenced by increased phosphorylation of PERK and IER1-α, which subsequently activate the TXNIP-dependent NLRP3 inflammasome/caspase-1 pathway, leading to increased IL-1B secretion, which ultimately leads to inflammation and apoptosis of endothelial cells in in vivo and in vitro studies [98]. Lu et al. also demonstrated that PA-induced ER stress was characterized by increased GRP-78 and CHOP expression, which impaired endothelium-dependent vasodilatation in rat thoracic aorta [99].

PA induced ER stress and activated unfolded protein response (UPR); meanwhile, ER stress causes endothelial–mesenchymal transition (EndMT), inhibits angiogenesis (cell migration, tube number), and decreases cell viability. However, PA-enhanced mesenchymal stem cells secreted stanniocalcin-1 (STC-1), which could attenuate ER stress-related EndMT, inhibition of angiogenesis, and cell viability in in vivo and in vitro studies [23].

Gustavo et al. disclosed that PA inhibits sarco/endoplasmic reticulum calcium ATPase (SERCA) expression and Ca^2+^ pump ATPase activity, which induces ER stress and activates JNK. Activated JNK blocks the vascular insulin pathway through decreased phosphorylation of Akt that results in insulin resistance [100].

#### Therapeutic Strategies Targeting on Ameliorate ER Stress

Ilexgenin A is a triterpenoid—a significant ingredient of ShanLv Cha. ShanLv Cha has been used to treat lipid disorders and cardiovascular diseases in China. Li showed that ilexgenin A suppresses PA-induced ER stress and TNXIP/NLRP3-related IL-1b secretion by activating AMPK. Meanwhile, ilexgenin A attenuated ROS generation and restored NO production and endothelial-dependent vasodilation in in vivo and in vitro studies; these studies suggested ilexgenin A could be developed as an inflammatory inhibitor for antagonized free fatty acid-induced endothelial dysfunction [98].

Fenofibrate (FF) is a fibric acid derivative that has been generally used for the treatment of hyperlipidemia for years [82]. Lu et al. used FF to reduce lipid excess deposits and elastic fiber thickening and maintain vascular endothelial integrity of rat thoracic aorta. Meanwhile, FF reversed endothelium-dependent vasodilation. Pretreatment with FF attenuated PA-induced ER stress and restored phosphorylation of eNOS and NO levels in an in vivo study [99]. These findings showed that fenofibrate can counteract macrovascular disease in the future.

Curcumin is produced from Curcuma longa; it possesses antioxidative, anti-inflammatory, and anti-cancer properties. PA enhanced expression of lectin-like oxidized low-density lipoprotein receptor-1 (LOX-1) through ER stress. Luo et al. showed that curcumin attenuated PA-induced ER stress and expression of LOX-1. Meanwhile, curcumin-restored PA inhibits angiogenesis and cell viability of endothelial cells [101]. These results indicated that curcumin has a preventive effect on anti-atherosclerosis.

The mechanisms targeted in ER stress are summarized in Figure 6.

### 2.7. Insulin Resistance

PA interfered with endothelial insulin signaling by increasing the phosphorylation of insulin receptor substrate-1 (IRS-1) serine and decreasing the phosphorylation of IRS-1 tyrosine to suppress the IRS-1/p-AKT/p-eNOS/NO pathway [45]. Fratantonio et al. found a similar result: PA increases phosphorylation of p-IRS at serine 307 and decreases NO generation, leading to insulin resistance of endothelial cells [102]. Moreover, PA-induced IKKβ disturbed insulin signaling by increasing IRS-1 serine phosphorylation and decreasing IRS-1 tyrosine phosphorylation to inhibit the AKT/eNOS/NO pathway [42]. In addition, upregulating sodium-glucose cotransporters 1 (SGLT1) and 2 (SGLT2) by PA suppresses the PI3K/AKT/eNOS signaling pathway, leading to reduced glucose consumption and NO production in vitro [103].

#### Therapeutic Strategies Targeting on Against Insulin Resistant

Clinopodium Chinense (CCE) has anti-inflammatory, antitumor, antioxidant, anti-cardiotoxicity, and antiradiation properties even though it was used to treat diabetes by the Fujian people of China [45]. A previous study showed that an extract of CCE reduced the high glucose-induced apoptosis of HUVEC by regulating the pAkt/NF-κB/Bax/Bcl-2/caspase-3 pathway [52]. Shi et al. revealed that pretreatment with ethyl acetate CCE inhibited TLR4, NF-kB, and AMPK expression, eliminated inflammation, and restored tyrosine phosphorylation and insulin-mediated vasodilation signaling pathways in in vivo and in vitro studies. This result showed that CCE has the potential to treat inflammation and insulin resistance induced by lipotoxicity in vascular endothelial cells [45].

Cyanidin-3-O-glucoside (C3G) is an anthocyanin that belongs to the flavonoid. Fratantonio et al. discovered that pretreatment with C3G was able to reduce ROS production and NF-kB nuclear translocation through Nrf2/EpRE-mediated gene expression like HO-1 and NQO1, which consequently attenuated leukocyte adhesion and meanwhile, promoted total antioxidant activity [104]. In addition, pretreatment with cyanidin-3-O-glucoside (C3G) could regulate phosphorylation of p-IRS and subsequently activate the Akt/eNOS pathway to restore NO production through inhibition of IKK and JNK, which is Nrf2 dependent [102]. These findings suggest that C3G can be a ROS scavenger and exert the ability to prevent insulin-resistant cardiovascular disease.

Adiponectin (APN) is an adipocyte-secreted hormone that reduces ROS generation through suppressed expression of NOX2 and follows the NF-kB pathway to diminish inflammatory cytokines (TNF-α, IL-6, ICAM-1). Meanwhile, pretreatment with APN restored the IRS-1/AKT/eNOS/NO signaling pathway to ameliorate PA-induced vascular insulin resistance [42]. Therefore, APNs exert cardiovascular protective effects against oxidative stress to improve endothelial dysfunction.

Phlorizin is a flavonoid and acted as an inhibitor of the sodium-glucose cotransporters (SGLTs); it ameliorated hyperglycemia and insulin resistance in an in vivo study [105]. Cotreatment with phlorizin reverses glucose consumption of endothelial cells and restores NO generation by inhibiting sodium-glucose cotransporter (SGLT1) and SGLT2 and subsequently activating the PI3K/Akt/eNOS pathway [103]. These results implied that phlorizin provided therapeutic options for patients with type II DM and vascular disease.

Glycyrrhiza glabra leaf is a medicinal plant. The main component of the extract from the leaves is polyphenol, but there are other metabolites, such as inositol D-pinitol. D-pinitol increased insulin sensitivity and ameliorated glycemic control in type II DM patients [106]. D-pinitol also reduced endothelial dysfunction, decreased ROS generation, and maintained NO signaling in an in vitro study [94]. Siracusa et al. disclosed that PA induced insulin-resistant-related endothelial dysfunction by suppressing the IRS/PI3K/AKT/eNOS pathway, but glycyrrhiza glabra leaf methanolic extract (GGLME) mainly contained D-pinitol, which promoted the expression of IRS, PI3K, AKT, and eNOS to protect the endothelial cell from lipotoxicity-associated metabolic disease. The protective effect of GGLME may come from pinitol contained in it, but GGLME is also rich in flavanones and dihydrostibenes, which are helpful in metabolic disease [107].

The mechanisms targeted in insulin resistance are summarized in Figure 7.

### 2.8. eNOS Pathway and NO Production

PA exposure increased NADPH oxidase (NOX) with enhanced NOX subunit gp91phox, p67phox, p47phox, and p22phox expression. Meanwhile, PA promoted protein phosphatase 2A (PP2A) and protein phosphatase 4 (PP4) expression, but PA reduced NO production and suppressed p-eNOS (Ser 633), p-eNOS (Ser 1177), and the PP4 regulatory subunit R2 (PP4R2) in endothelial cells. Silencing of PP2A and PP4 restored PA-inhibited p-eNOS (Ser 1177) and p-eNOS (Ser 633) expression. Also, silencing of the NOX subunit p67phox improved expression of p-eNOS (Ser 633) and PP4R2. Additionally, overexpression of PP4R2 increase NO production, p-eNOS (Ser 633) expression, and improved wound healing and tube formation, which were impaired by PA. These findings suggested that PA decreased NO production of endothelial cells through suppressing PP4R2, which activates PP4 to inhibit p-eNOS (Ser 633), and PA also enhances PP2A to inhibit p-eNOS (Ser 1177) expression [108].

#### Therapeutic Strategies Targeting on Ameliorate eNOS Pathway and NO Production

Icariside II (ICA II) is an active flavonoid and the primary pharmacological metabolite of icariin extracted from the traditional Chinese medicine Epimedium. ICA II can improve endothelial dysfunction through regulating MAPK and Akt-eNOS signaling pathways from previous studies. Gu et al. revealed that ICA II restored NO production via enhancing the PA-impaired SRPK1-Akt-eNOS signaling pathway and cellular viability. ICA II may be a potential drug for treating diseases associated with endothelial dysfunction to maintain human health [109].

Leflunomide is an antirheumatic drug. Jiang et al. demonstrated that cotreatment with leflunomide or its active metabolite teriflunomide with PA could enhance NO production, p-eNOS, and p-AMPKα by inhibition of dihydroorotate dehydrogenase (DHODH) to ameliorate endothelial dysfunction. Additionally, leflunomide or teriflunomide reduces lipid accumulation in alpha mouse liver 12 (AML12) cells via DHODH/AMPK signaling pathways. In an in vivo study, leflunomide exerted an anti-atherosclerotic effect by diminishing total cholesterol and triglyceride in plasma and liver, reducing plasma sugar levels, and attenuating atherosclerotic plaque area [110]. These findings provided a promising drug for preventing cardiovascular disease and treatment of atherosclerosis.

Ginsenoside Rh4 is a compound exclusively in Panax ginseng that exhibited antitumor properties in a previous study. Zhang et al. disclosed that pretreatment with Rh4 increased cell viability, enhanced phosphorylation of AMPK, and downstream targets of AMPK like PGC-1α, complexes I, III, IV, and V of endothelial cells, which PA impaired. Rh4 also restored p-eNOS expression and NO level and diminished P38/NFкB-mediated inflammation that was induced by PA. Rh4 also maintains the mitochondrial function of endothelial cells via promoted ATP production and spare respiratory capacity. In vivo study, Rh4 reduced serum cholesterol, triglyceride, and ROS levels in P407-induced hyperlipidemia mice. Furthermore, Rh4 reversed NO level, p-AMPK, p-Akt, and p-eNOS expression in the thoracic endothelium of hyperlipidemic mice [111]. These results reveal ginsenoside Rh4 can prevent endothelial injury induced by hyperlipidemia by activating the AMPK signaling pathway.

### 2.9. Long Non Coding RNA/miRNA

Moran et al. used transcriptomic analysis to demonstrate that 64 long non-coding RNAs (lncRNAs) were upregulated and 46 were downregulated after PA treatment. Colorectal neoplasia differentially expressed (CRNDE) is one of the lncRNAs induced by PA. Knockdown of CRNDE by siRNA inhibits cell migration and tube formation. In addition, the knockdown of CRNDE suppresses the G1 phase cell cycle by promoting the expression of p21. However, CRNDE is not directly regulated by PA; it may compensate to maintain angiogenesis after PA treatment. AGE-RAGE signaling pathway in diabetic complications, IL-17 signaling, and cysteine and methionine metabolism are new findings in the KEGG pathway. Meanwhile, ferroptosis, HIF-1 signaling, ER stress, and ER stress-related apoptosis are enriched in the KEGG pathway and gene ontology, respectively; thereby, the possible connection between palmitic acid-induced endoplasmic reticulum stress and CRNDE induction needs to be examined in future studies, providing a potential mechanism by which CRNDE expression is induced after palmitic acid [112].

Gong et al. discovered that PA enhanced long non-coding RNA (lncRNA) hypoxia-inducible factor 1α-antisense RNA 1 (HIF1A-AS1) expression, which promotes apoptosis and impaired cellular migration. Overexpression of HIF1A-AS1 promoted miR-1298-5p, miR-30c-5p, and miR-27b-5p expression and downregulated miR-4664-3p, miR-769-5p, miR-106b-5p, and miR-548o-3p expression; this miRNA may correlate with the endothelial cell apoptotic process, cellular carbohydrate metabolic process, or type I interferon signaling pathway [113].

PA increases ROS production, apoptosis, and proinflammatory cytokines (IL-6, TNF-α) and inhibits cell migration through Wnt/B-catenin by suppressing the expression of miR-155. Overexpression of miR-155 attenuated apoptosis, ROS production, and inflammation and enhanced cell migration by regulating the Wnt/B-catenin pathway. Therefore, modulating miR-155 can potentially treat palmitic acid-induced vascular endothelial injury [114].

### 2.10. Other Mechanisms

PA disturbed endothelial cell angiogenesis by inhibiting tube formation through suppressed PI3K/Akt/Foxo-1/GTPCH1 signaling pathway and NO production [78]. PA upregulated expression of endothelin-1 (ET-1) after acute and chronic exposure to PA, which was regulated by ER stress and PKC in in vitro and in vivo studies. ET-1 may be attributed to obesity-related cardiovascular dysfunction [115].

C1q/tumor necrosis factor-related protein 13 (CTRP13) is a member of the adipokine family, and it has been reported that it could inhibit the progression of atherosclerosis. Zhu et al. discovered that PA suppressed the expression of CTRP13. Still, overexpression of CTRP13 can upregulate NO levels and tube formation through AMPK-regulated NOX1/p38 and KLF2 and reduce apoptosis, ROS production, and inflammation in an in vitro study [116].

IL-33 has a potent immunomodulatory and anti-atherosclerosis function, which belongs to the IL-1 family from previous studies. Hosomi et al. revealed that IL-33 upregulated fatty acid metabolism-related genes like SCD, SREBF1, and FASN to decrease saturated fatty acid accumulation induced by PA in HAEC via promoting the synthesis of saturated to unsaturated fatty acids. IL-33 receptor knockout mice (APOEST2 DKO mice) fed with a high-fat diet also showed increased levels of saturated fatty acid (myristic acid, palmitic acid, stearic acid), inflammatory cytokine-related genes (Mcp1, Ccl2, IL-1b, Ifng, and Tnfa), increased M1/M2 macrophage ratio, and decreased type 2 innate lymphocytes (ILC2), which exhibit anti-inflammatory and anti-atherosclerosis properties in the aorta region [117]. These results demonstrated that IL-33 is an activator of ILC2; it can potentially reduce atherosclerosis through innate immunity.

Accumulation studies proved that PA disturbed endothelial cells’ function through different signaling pathways. Still, Tan et al. revealed that accumulation of lipid droplets in endothelial cells or aorta-induced ciliary loss of endothelial cells contributes to atherosclerosis progression, which is related to decreasing PA levels. Feed with a PA-enriched diet or using stearoyl-CoA desaturase 1 (SCD1) inhibitor to enhance PA levels, which reduces endothelial ciliary loss, maintains ciliary homeostasis, and attenuates atherosclerosis progression in vivo study. These results implicated PA levels playing a critical role in lipid droplet accumulation and atherosclerosis [118].

Table 1 and Table 2 summarize recent studies of fatty acid-induced vascular endothelial dysfunction and different targeting strategies to counteract vascular lipotoxicity.

## 3. Future Perspective

From this review, we know that clinical application drugs such as dapagliflozin (DAPA), canagliflozin (CAN), and leflunomide can improve endothelial dysfunction caused by free fatty acid PA. In the future, more clinical drugs can be discovered to prevent or treat cardiovascular diseases caused by hyperlipidemia. There are relatively few epigenetic studies on endothelial dysfunction caused by PA, and research can be conducted in related aspects.

## 4. Conclusions

Saturated fatty acid, especially palmitic acid (PA), induced vascular dysfunction through increased ROS generation, induced inflammation, ER stress, apoptosis, necroptosis, and even pyroptosis of endothelial cells. Moreover, PA could activate autophagy but disturb autophagy flux. Moreover, PA impairs mitochondrial function and induces insulin resistance. However, different strategies and efforts have been tried to alleviate PA-induced vascular lipotoxicity by using unsaturated fatty acids, flavonoids, and traditional Chinese medicine to reduce ROS production, attenuate insulin resistance, restore NO production, and diminish inflammation of vascular endothelial cells to preserve endothelial function that would reduce cardiovascular risk, which impacts human health.

## Figures and Tables

**Figure 1 antioxidants-13-01486-f001:**
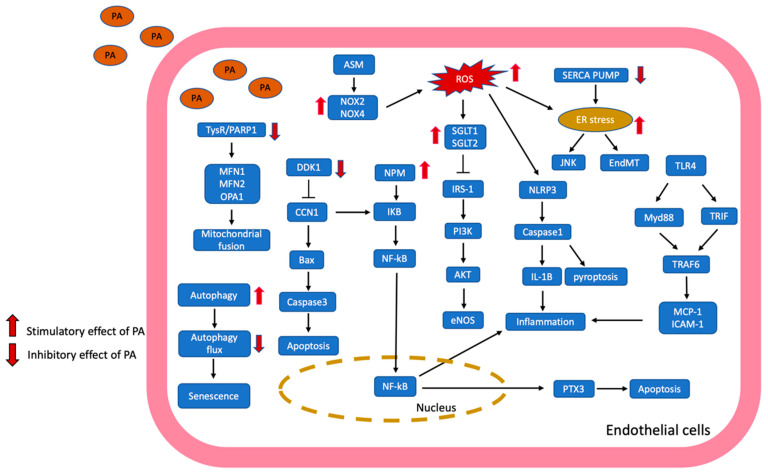
The potential molecular pathways of fatty acid, especially PA-induced vascular endothelial dysfunction, are illustrated and summarized—PA uptake by endothelial cells through fatty acid transporter or passive diffusion. PA activated ASM to enhance ROS generation through NOX2 and NOX4. Moreover, PA inhibited the SERCA pump, which induced ER stress and increased JNK expression. ER stress induced by PA promoted endothelial mesenchymal transition (EndMT). PA also inhibited the TysR/PARP1 pathway to downregulate the expression of MFN1, MFN2, and OPA1, leading to the fragmentation of mitochondria. PA could activate autophagy but suppress autophagy flux, which may modulate the senescence of endothelial cells. Furthermore, PA increases NPM levels to activate the NF-kB pathway, leading to inflammation. Meanwhile, PA started the TLR4-dependent MyD88 and TIRF pathway, growing the inflammatory mediator MCP-1 and ICAM-1 levels. PA suppressed DDK1 to induce CCN1-related cell apoptosis. In addition, PA activated the NLRP3/caspase-1 pathway to induce pyroptosis of endothelial cells. Moreover, PA enhanced the expression of SGLT1 and SGLT2 to inhibit the PI3K/AKT/eNOS signaling pathway, leading to insulin resistance. ASM: acid sphingomyelinase; Bax: Bcl-2 associated x protein; CCN1: cysteine-rich angiogenic inducer 61; DDK1: Dickkopf-1; ER stress: endoplasmic reticulum stress; ICAM-1: intercellular cell adhesion molecule-1; IKB: inhibitor of nuclear kappa B protein; IRS-1: insulin receptor substrate-1; MCP-1: monocyte chemotactic protein-1; MFN1: mitofusin-1; MFN2: mitofusin-2; MyD88: myeloid differentiation primary response gene 88; NPM: nucleophosmin; NOX2: NAPDH oxidase 2; NOX4: NAPDH oxidase 4; NLRP3: NOD-like receptor protein 3; OPA1: optic atrophy 1 protein; PTX3: pentraxin 3; SGLT1: sodium glucose cotransporters 1; SGLT2: sodium glucose cotransporters 2; SERCA: sarco/endoplasmic reticulum calcium ATPase; TLR4: toll-like receptor 4; TRIF: toll/interleukin-1 receptor (TIR)-domain containing adaptor-inducing interferon-b; TRAF6: tumor necrosis factors (TNF) receptor-associated factor-6; TysR/PARP1: tyrosyl transfer-RNA synthetase/poly (ADP-ribose) polymerase 1.

**Figure 2 antioxidants-13-01486-f002:**
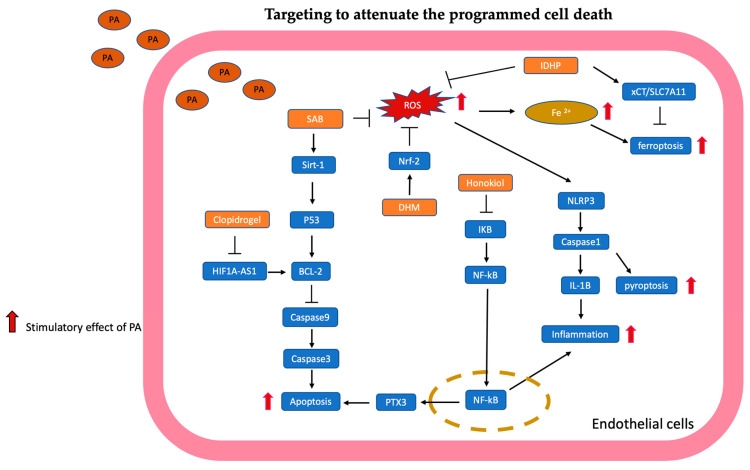
Targeting to attenuate the programmed cell death is illustrated and summarized. Salvianolic acid B (SAB) upregulates the expression of Sirt-1 to reduce apoptosis through the BCL-2 pathway. Honokiol suppressed PTX3 expression through the NF-kB pathway to attenuate apoptosis. Dihydromyricetin (DHM) enhances the Nrf-2 signaling pathway to reduce ROS production, subsequently inhibiting NLRP3-related pyroptosis. Clopidogrel inhibited long non-coding RNA HIF1A-AS1 to reduce apoptosis by BCL-2/caspase-9/caspase-3 pathway. Isopropyl 3-(3,4-dihydroxyphenyl)-2-hydroxypropionate (IDHP) reduced ROS to decrease intracellular ferrous ion levels, which are involved in ferroptosis; meanwhile, IDHP promoted xCT/SLC7A11 expression to suppress ferroptosis of endothelial cells.

**Figure 3 antioxidants-13-01486-f003:**
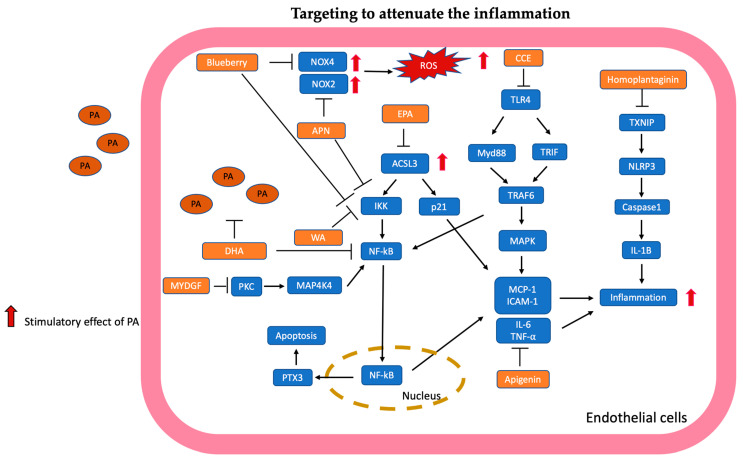
Targeting to attenuate the inflammation is illustrated and summarized. Eicosapentaenoic acid (EPA) suppresses long-chain acyl-CoA synthetase (ACSL) to reduce inflammation via p21 and NF-kB signaling pathways. Docosahexaenoic acid (DHA) reduces intracellular PA accumulation, and DHA also attenuates inflammation through the NF-kB pathway. Adiponectin (APN) inhibits NOX2 expression to suppress ROS production, and APN inhibits the NF-kB pathway to reduce the inflammation of endothelial cells. Blueberries inhibit NOX4 expression to reduce ROS production. Meanwhile, blueberries suppress NF-kB pathway-related inflammation. Clinopodium Chinense (CCE) inhibits the expression of toll-like receptor 4 (TLR4), which consequently suppresses MyD88/TRIF/TRAF 6 to reduce inflammation with inhibition of MAPK and NF-kB pathway. Homoplantaginin suppressed TXNIP/NLRP3/caspase-1-related pathway to reduce inflammation. Apigenin suppressed proinflammatory cytokines IL-6, TNF-α, and adhesion molecules ICAM and VCAM related to inflammation. Withaferin A (WA) inhibited inflammation through NF-kB-related pathways. Myeloid cell-derived growth factor (MYDGF) suppressed PKC/MAP4K4/NF-kB signaling pathway to ameliorate inflammation of endothelial cells.

**Figure 4 antioxidants-13-01486-f004:**
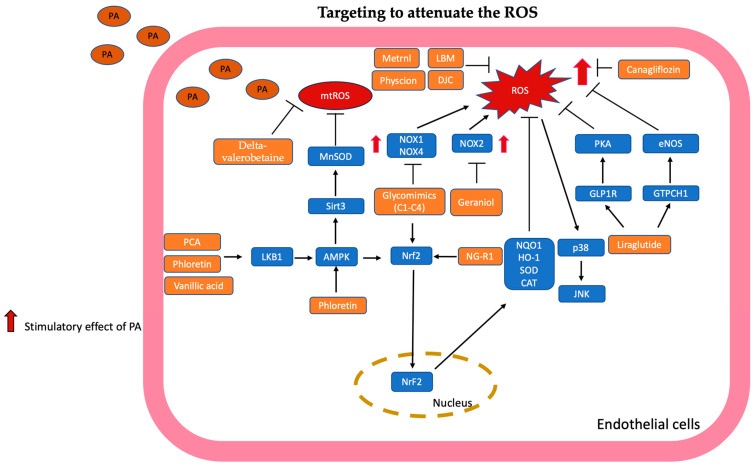
Targeting to attenuate the ROS is illustrated and summarized. Glycomimetics (C1–C4) suppressed NOX1 and NOX4 expression and enhanced the Nrf2 pathway to reduce ROS production. Phloretin activated the LKB1/AMPK/Nrf2 pathway to reduce ROS production. In addition, phloretin increases the activity of MnSOD through AMPK/Sirt-3 to decrease ROS accumulation. Protocatechuic acid (PCA) and vanillic acid could activate the LKB1/AMPK/Nrf2 pathway to reduce ROS generation. Liraglutide promoted GLP-1R/PKA and GTPCH1/eNOS signaling pathways to inhibit ROS production. Loliolus beka gray meat (LBM) extract, Danzhi jiangtang capsule (DJC), meteorin-like protein (Metrnl), and physcion could reduce the ROS production directly. Geraniol (GOH) downregulated NOX2-dependent ROS generation. Delta-valerobetaine (δVB) could diminish mitochondrial ROS (mtROS). Canagliflozin (CAN) attenuated excess ROS production in endothelial cells. Notoginsenoside-R1 (NG-R1) activated the Nrf2 pathway to reduce ROS production.

**Figure 5 antioxidants-13-01486-f005:**
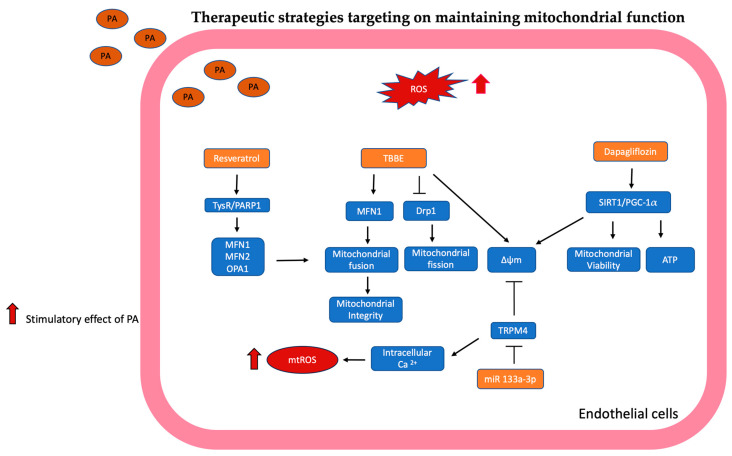
Targeting to maintain mitochondrial function is illustrated and summarized. DCI-enriched Tartary buckwheat bran extract (TBBE) inhibits Drp1-related mitochondrial fission; TBBE also enhances MFN1 expression-related mitochondrial fusion and maintains mitochondrial integrity and mitochondrial membrane potential (Δψm). Resveratrol (RSV) activated the TysR/PARP1 signaling pathway to reverse the expression of MFN1, MFN2, and OPA1, sequentially maintaining mitochondrial morphology and mitochondrial membrane potential to preserve endothelial function. Dapagliflozin (DAPA) activated the SIRT1/PGC-1α pathway to increase mitochondrial viability, ATP production, and mitochondrial biogenesis. miR 133a-3p inhibits TRPM4 expression to restore mitochondrial membrane potential and reduce intracellular calcium accumulation-related mitochondrial ROS production.

**Figure 6 antioxidants-13-01486-f006:**
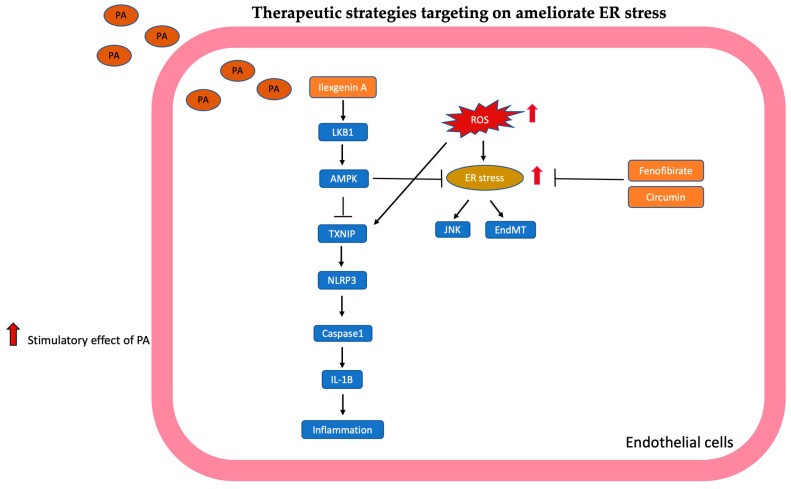
Targeting to ameliorate ER stress is illustrated and summarized. Ilexgenin A suppresses ER stress and TXNIP/NLRP3-related inflammation by activating AMPK. Fenofibrate (FF) and curcumin attenuated PA-induced ER stress.

**Figure 7 antioxidants-13-01486-f007:**
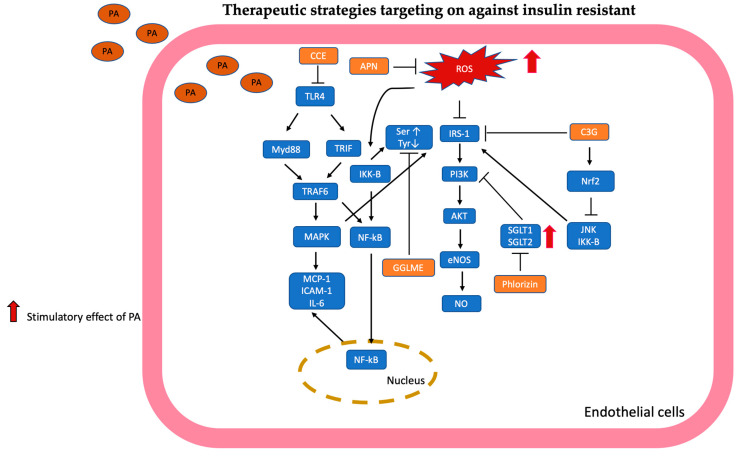
Targeting to insulin resistance is illustrated and summarized. Clinopodium Chinense (CCE) inhibits the expression of toll-like receptor 4 (TLR4), which consequently suppresses MyD88/TRIF/TRAF 6 to affect phosphorylation of IRS-1 through inhibition of MAPK and NF-kB pathway. Cyanidin-3-O-glucoside (C3G) enhanced Nrf2 expression to suppress IKK and JNK, subsequently regulating phosphorylation of p-IRS to activate the Akt/eNOS pathway to restore NO production. Adiponectin (APN) inhibited the ROS/IKK-B pathway to modulate phosphorylation of IRS-1, which attenuated insulin resistance. Phlorizin inhibited SGLT1 and SGLT2 expression to activate the PI3K/Akt/eNOS pathway to restore NO generation. Glycyrrhiza glabra leaf methanolic extract (GGLME) restores phosphorylation of IRS-1 to activate the PI3K/Akt/eNOS pathway.

**Table 1 antioxidants-13-01486-t001:** In vivo and in vitro fatty acid-induced endothelial dysfunction.

Animal Mode	Cell Type	Time	Dose	Result	References
Apoe knockout mice and Ldlr knockout mice; HFD for 12 weeks	HUVEC	24 h	PA 0.3 mM	IQGAP1 increased ROS of mitochondrial and released mtDNA to cytoplasm, which activated cGAS-STING-related pathway to induce pyroptosis and inflammation	An et al. [93]
Apoe knockout mice; HFD for 8 weeks	HUVEC	24 h	PA 0.2 mM	ANGPTL4 regulated PA impaired cell viability through autophagy	Zhan et al. [63]
	HUVEC, RCCEC	24 h	PA 0.25 mM	PA inhibited cysteine/GSH/GPX4 axis and increased intracellular Fe^2+^ accumulation, which triggered ferroptosis	Tan et al. [31]
Lipid emulsion induced coronary microvascular dysfunction (CMD) model, C57BL/6	CMECs	24 h	PA 400 μM	Restoration of AMPK/KLF2/eNOS signaling pathway decreased ROS production, leukocyte adhesion, coronary flow	Zhang et al. [71]
	HUVEC	24 h	PA 200 μM	PA increased IL-1β, CCL3, CCL4, SPP1, IRF5, MMP7, and MMP9 levels, which correlate to carotid atherosclerosis	Zhang et al. [17]
	HUVEC	24 h	PA 500 μM	PA induced cellular senescence through ROS production	Zhu et al. [16]
	HUVEC	36 h	PA 0.4 mM	PA enhanced HIFIA-AS1 expression that promoted apoptosis and disturb cellular migration	Gong et al. [113]
	HUVEC	24 h	PA 0.2, 0.4, 0.8 mM	PA induced apoptosis and inflammation through DDK1 regulated CCN1 and Wnt/B-catenin	Gan et al. [24]
ATF4 deficiency mice HFD vs. chow diet for 20 weeks	HCMECs	24 h	PA 50–400 μM	AT4 maintained H3K4 methylation of NOS3, ERK1 through MAT2A and MLL1	Chen et al. [25]
	HUVEC	24 h	PA 300, 500 μM	PA induced expression of MAFG and MAFF which inhibit cell apoptosis	Wang et al. [26]
APOE knockout mice HFD vs. chow diet for 16 weeks	HUVEC	24 h	PA 500 μM	NPM increased atherosclerotic plaque, activated NF-kB and downstream genes, leading to endothelial inflammation	Rao et al. [44]
Myeloid cell-derived MYDGF knockout mice HFD vs. chow diet for 12 weeks	MAEC	24 h	PA 0.4 mM	MYDGF attenuated HFD or PA-induced apoptosis, inflammation, and increased adhesion molecules (VCAM-1, ICAM-1, E-selectin) with PKC/MA4K4/NF-kB pathway	Meng et al. [57]
	HAEC	6 h	PA 200 μM	PA induced proinflammatory response, monocyte adhesion, and insulin-related vasodilation through ACSL	Ren et al. [47]
	HUVEC	24 h	PA 0.3 mM	PA induced autophagy promoting ROS production through Ca^2+^/PKCα/NOX4 pathway, leading to endothelial dysfunction	Chen et al. [58]
SD rats, CB7BL6 mice	HUVEC	24 h	PA 100 μM	PA impaired K(Ca)2.3 channel-related vasodilation through NOX/ROS/p-38/NF-kB pathway	Wang et al. [66]
	HUVEC	48 h	PA 450 μM/L	PA induced apoptosis of endothelial cell through suppressing k-OR regulated PI3K/AKT/eNOS signaling pathway	Cui et al. [119]
	HUVEC, HAEC	12 h	PA 100 μM	PA induced expression of CRNDE, which regulated angiogenesis and cell cycle of endothelial cell	Moran et al. [112]
	HUVEC	24 h	PA 100 μM	PA suppressed expression of miR-155 leading to increased apoptosis, ROS production, proinflammatory cytokine (IL-6, TNF-a), and decreased cell migration	Zhao et al. [114]
SD rats; 10 weeks HFD	RAEC	48 h	PA 100 μM	PA enhanced expression of ASM to suppress IRS/PI3K/AKT/eNOS signaling pathway through NOX2, NOX4-mediated superoxide production	Li et al. [21]
C57BL/6 mice; 16 weeks HFD	HAEC	24 h (PA 100 μM)	12–16 days (PA 20 μM)	PA upregulated expression of ET-1 through ER stress and PKC	Zhang et al. [115]
	HAEC	24 h	PA 0.3 mM	PA reduced mitochondrial membrane potential and induced mitochondrial damage. cCAMP activated STING/IRF3/MST1 signaling pathway to promote phosphorylation of YAP l to suppress angiogenesis	Yuan et al. [90]
	HAEC	24 h	PA 200 μM, 300 μM, 400 μM, 500 μM	PA activated cGAS-STING pathway to induce endothelial-to-mesenchymal transition	Liu et al. [91]
Sting deficiency mice (C57BL/6); HFD 12 weeks	HAEC	24 h	PA 0.3 mM	PA induced endothelial inflammation by activating cGAS/STING/IRE3 pathway. HFD-fed STING deficiency mice reduced macrophage infiltration and vascular ICAM-1 expression	Mao et al. [48]
MD2 knockdown mice HFD 40 weeks	RAEC	6 h	PA 0.15 mM	PA induced ROS generation and vascular fibrosis through (MD2) by suppressed AMPK/Nrf-2 pathway in in vivo and in vitro studies	Wang et al. [70]
	HUVEC	36 h	PA 100 μM	PA inhibited NO production through NOX related PP2A and PP4 to suppress phosphorylation of eNOS at Ser633 and Ser1177	Liang et al. [108]
	HUVEC	24 h	PA 0.25~1 mM	PA suppressed SERCA pump resulting in ER stress, which consequently activated JNK to inhibit AKT, leading to vascular insulin resistance	Gustavo et al. [100]
	HUVEC	24 h	PA 200 μM	PA suppressed CTRP13 expression, overexpression of CTRP13 reduced ROS production, inflammation, cellular apoptosis through AMPK-related KLF2, NOX1/p38 pathway	Zhu et al. [116]

HAEC: human artery endothelial cell; HFD: high fat diet; HUVEC: human umbilical vein endothelial cell; MAEC: mouse artery endothelial cell; RAEC: rat artery endothelial cell; SD: Sprague-Dawley rat.

**Table 2 antioxidants-13-01486-t002:** Prevention of fatty acid-induced vascular endothelial dysfunction.

Protective Agent	Animal Model	Cell Type	PA Dose	Result/Conclusion	References
CTRP9		HUVEC	PA 0.5 mM	CTRP9 ameliorated PA induced senescence through AMPK-regulated autophagy	Lee et al. [60]
Dapagliflozin (DAPA)	HFD + DAPA (1 mg/kg/day, intragastric) for 16 weeks	HUVEC	PA 200 μM	DAPA activated SIRT1/PGC-1a to restore mitochondrial membrane potential, mitochondrial viability, ATP production, mitochondrial biogenesis, which was impaired by PA	He et al. [95]
Selegiline	Apoe knockout mice, HFD 16 weeks, selegiline 0.6 mg/kg/day, IP for 6 weeks	HUVEC	PA 500 μM	Selegiline reduced ROS production, proinflammatory cytokine, apoptosis and atherosclerotic lesion through suppressing MAOB	Tian et al. [72]
Clopidogrel		HUVEC	PA 0.4 mM	Clopidogrel decreased apoptosis through inhibition of HIF1A-AS1 expression	Wang et al. [35]
Icariside II (ICA II)		HUVEC	PA 150 μM	ICA II increased SRPK1-Akt-eNOS pathway to restore NO production	Gu et al. [109]
δ-valerobetaine (δVB)		TeloHAEC	PA 0.5 mM	δVB restored SIRT3 expression to attenuate ROS production, inflammation, pyroptosis, and insulin resistance	Martino et al. [30]
Canagliflozin (CAN)		HUVEC	PA 300 μM	CAN decreased cell senescence, DNA damage, cell cycle arrest through suppressing ROS-dependent p38/JNK pathway	Hao et al. [84]
Docosahexaenoic acid (DHA)		HUVEC	PA 1 mM	DHA decreased deposition of lipid, inflammatory cytokine, apoptosis and increased cell survival rate after PA treatment	Abriz et al. [51]
DHA		HUVEC	PA 0.1 mM	DHA promoted miR-3691-5p to suppress caspase-3 and serpine1 related apoptosis	Wu et al. [37]
DHA		HUVEC	PA 1 mM	DHA diminished inflammatory cytokine, apoptosis, and intracellular lipid accumulation	Novinbahador et al. [120]
Apigenin (Api)		HAEC	PA 150 μM	Api attenuated expression of cell adhesion molecules, inhibited proinflammatory cytokines, enhanced FOXO1 expression, and increased cell viability after PA treatment	Miao et al. [4]
Mesenchymal stem cell (MSC)	SD rats HFD 16 weeks; 2 × 10^6^ MSC/mL, IV, 8 times/week	HUVEC	PA 150 μM	MSC secreted STC-1 ameliorated PA-induced ER stress-mediated EndMT, angiogenesis, and cell viability	Luo et al. [23]
Curcumin		HUVEC	PA 100 μM	Curcumin restored PA inhibit angiogenesis and cell viability, alleviated PA-induced ER stress and expression of LOX-1	Luo et al. [101]
Liraglutide (Lira)		MS-1	PA 0.25 mmol/L	Lira inhibited apoptosis and ROS production through GLP-1R/PKA and GTPCH1/eNOS signaling pathways after PA treatment	Le et al. [79]
Phloretin	C57BL/6J HFD12 weeks;	HUVEC	PA100 μM	Phloretin restored expression of MnSOD by AMPK/Sirt3 to decrease mtROS production	Han et al. [74]
Phloretin		HUVEC	PA 0.1 mM	Phloretin attenuated PA induced oxidative stress through LKB1/AMPK/Nrf-2 pathway by promoting SOD, GPx expression and restored mitochondrial membrane potential	Yang et al. [67]
Phloretin		HUVEC	PA100 μM	Phloretin inhibited LncBAG6-AS to restore total antioxidant capacity, glutathione reductase activity, which attenuated oxidative stress induced by PA	Li et al. [75]
Methanolic extract from liquorice leaves (GGLME)		HUVEC	PA100 μM	GGLME restored PA inhibited IRS1/PI3K/AKT/eNOS signaling pathway	Siracusa et al. [107]
Ethyl acetate extract of *Clinopodium Chinense* (CCE)		HUVEC	PA 100 umol/L	CCE inhibited TLR4/NFKB/MAPK to reduce inflammation and restore insulin-mediated vasodilation signaling pathway after PA exposure	Shi et al. [45]
Resveratrol (RSV)		HUVEC	PA 0.2 mM	RSV activated TysR/PARP signaling pathway to suppress ROS generation, which reversed expression of MFN1, MFN2, and OPA1 to maintain mitochondrial morphology	Yang et al. [22]
RSV	C57BL/6; HFD + RSV (20 mg/kg/day) oral feeding	HAEC	PA 0.4 mmol/L	RSV promoted AMPK/mTOR-dependent autophagy to attenuate ROS production and enhance p-eNOS expression	Song et al. [62]
Extract of LBM		HUVEC	PA 0.5 mM	Extract of LBM attenuated PA induced ROS production and proinflammatory cytokine	Lee et al. [80]
Salvianolic acid B (SAB)		HUVEC	PA 0.75 mM	SAB attenuated p53-related apoptosis by enhancing expression of Sirt-1 and BCL-2. SAB reduced ROS generation and TXNIP expression, reversing mitochondrial membrane potential	Zhai et al. [32]
Blueberry metabolite		HAEC	PA 0.5 mM	Blueberry metabolite inhibited NOX4 activity and ROS production, restored NO production to diminish PA induce vascular dysfunction	Bharat et al. [43]
Dihydromyricetin (DHM)		HUVEC	PA 0.3 mM	DHM attenuated pyroptosis is NRF2 dependent, DHM reduced intracellular ROS and mitochondrial ROS to decrease LDH and IL-1b	Hu et al. [29]
Danzhi jiangtang capsule (DJC)	SD rats HFD 12 weeks + DJC (500, 1000 mg/kg/day) 8 weeks	HUVEC	PA 0.5 mM	DJC attenuated ER stress, oxidative stress, promoted antioxidant, decreased inflammatory cytokine and apoptosis	Lu et al. [81]
DCI-enriched Tartary buckwheat bran extract (TBBE)	SD rats, ICR mice HFD + TBBE (74.67 mg/kg, 149.3 mg/kg	RAEC	PA 0.1 mM	TBBE suppressed oxidative stress, ER stress, and JNK-associated inflammation, TBBE protected mitochondrial integrity through modulating phosphorylation of Drp1 and increased mitochondrial membrane potential	Zhang et al. [92]
Protocatechuic acid (PCA)		HUVEC	PA 0.1 mM	PCA suppressed PA-induced oxidative stress though activating Nrf-2 and PGC-1α-dependent AMPK	Han et al. [68]
Phlorizin		HUVEC	PA 0.3 mM	Phlorizin reversed NO production through suppressing PA-induced SGLT1 and SGLT2 expression and activating AKT/eNOS pathway	Li et al. [103]
Oleate (Ole) and palmitoleate (PO)		HUVEC	PA 0.25, 0.5 mM	Ole, PO attenuated PA-induced lipoapoptosis, cellular adhesion molecular (ICAM, E-selectin) through AMPK pathway	Lee et al. [27]
Vanillic acid (VA)		HUVEC	PA 0.1 mM	VA reduced production of ROS through activating LKB1/AMPK/Nrf-2 pathway, which promoted antioxidant enzyme activities and reversed mitochondrial membrane potential by regulating SIRT-1/PGC1-α	Ma et al. [82]
Cyanidin-3-O-glucoside (C3G)		HUVEC	PA 0.1 mM	Pretreatment with C3G regulated phosphorylation of p-IRS and subsequently activated Akt/eNOS pathway to restore NO production through inhibition of IKK and JNK is Nrf2 dependent.	Fratantonio et al. [102]
Glycomimics (C1–C4)		HUVEC	PA 0.1 mM	Pretreatment or cotreatment with glycomimics (C1~C4) inhibited NAPDH oxidase activity, which reduced ROS generation. Glycomimics promoted Nrf2-dependent antioxidant enzymes and restored NO production to dimmish oxidative damage	Mahmoud et al. [65]
Metformin + liraglutide		HUVEC	PA 0.5 mM	Cotreatment with metformin and liraglutide protected endothelial cell from lipotoxicity through GLP-1R and PKA signaling pathway	Ke et al. [69]
Adiponectin (APN)		HUVEC	PA 0.2 mM	Pretreatment with APN attenuated vascular inflammation and insulin resistance through inhibiting NOX2-dependent ROS/IKKb signaling pathway	Zhao et al. [42]
Homoplantaginin		HUVEC	PA 0.1 mM	Pretreatment with homoplantaginin attenuated PA-induced vascular inflammation by suppressing inflammatory cytokines (MCP-1, ICAM-1, IL-1B) through inhibition of TLR4 and NLRP3 signaling pathway and restoring NO generation	He et al. [46]
Geraniol (GOH)	C57BL/6 mice HFD for 8 + 6 weeks; GOH IP (100 mg/kg/day) for 6 weeks	HUVEC	PA 0.3 mM	GOH improved acetylcholine-dependent vascular relaxation, downregulated NOX2 to decrease ROS generation, suppressed adhesion molecular (ICAM-1, VCAM-1) to attenuate vascular inflammation	Wang et al. [83]
Liraglutide (Lira)		HUVEC	PA 0.5 mM	Cotreatment with Lira recover PA impaired endothelial tube by enhancing PI3K/Akt/Foxo-1/GTPCH1 signaling pathway and NO production	Ke et al. [69]
Withaferin (WA)		HUVEC	PA 0.1 mM	Pretreatment with WA abolished PA-induced vascular inflammation, insulin resistance and ROS generation through Ikkb/NF-KB and IRS/AKT/eNOS signaling pathway	Batumalaie et al. [56]
Eicosapentaenoic acid (EPA)		HUVEC	PA 100 μM	Cotreatment with EPA reduced ACSL to inhibit inflammatory-related cytokines, proteins, and adhesion molecules through suppressing NF-kB and P21	Ishida et al. [41]
Fenofibrate (FF)	SD rats; HFD + FF (30 mg/kg/day)	MAEC	PA 0.5 mM	FF attenuated ER stress, restored NO level, and decreased lipid accumulation, elastic fiber thickening of vascular	Lu et al. [99]
Ilexgenin A	ICR mice; HFD + ilexgenin A (80 mg/kg/day) oral gavage	EA hy926	PA 0.1 mM	Ilexgenin A attenuated ER stress through activation of AMPK to inhibit TNXIP/NLRP3 pathway-related inflammation and apoptosis; ilexgenin A attenuated ROS generation, restored NO production by enhancing p-eNOS expression	Li et al. [98]
Honokiol		HUVEC	PA 0.5 mM	Honokiol attenuated apoptosis and restored NO production through inhibiting NF-kb pathway-dependent PTX3	Qiu et al. [34]
Ginsenoside Rh4	P407-induced hyperlipidemia mice + Rh4 (50 mg/kg/day or 20 mg/kg/day) intraperitoneal injection	HAEC	PA 500 μM	Rh4 promoted, AMPK to reverse NO level, p-eNOS expression, maintain mitochondrial function, and reduce inflammation in hyperlipidemia-induced vascular injury	Zhang et al. [111]
Leflunomide	ApoE -/- mice; WD + leflunomide (20 mg/kg/day) oral gavage	HUVEC AML12	PA 500 μM	Leflunomide enhanced NO production through DHODH/AMPK/eNOS pathway; leflunomide attenuated atherosclerosis by lowering total cholesterol, triglyceride, plasma glucose	Jiang et al. [110]
IDHP		HUVEC	PA 0.3 mM	IDHP reduced ROS production, delay senescence through inhibition of ferroptosis	He et al. [40]
Physcion	SD rat; HFD + physcion (high: 50 mg/kg/day; low: 30 mg/kg/day) intragastric 12 weeks	HUVEC	PA 0.25 mM	Physcion reduced apoptosis, ROS production, inflammation, ER stress; physcion also restored p-Akt, p-eNOS, NO, SOD, Nrf-2 expression	Wang et al. [88]
Notoginsenoside (NG)-R1		HUVEC	PA 100 μM	NG-R1 restored glucose uptake, decrease insulin resistance and ROS production through Nrf2-related pathway	Wang et al. [87]

HAEC: human artery endothelial cell; HFD: high fat diet; HUVEC: human umbilical vein endothelial cell; MAEC: mouse artery endothelial cell; MS-1: immortal mouse islet microvascular endothelial cell line; RAEC: rat artery endothelial cell; SD: Sprague-Dawley; ICR: Institute of Cancer Research; AML12: alpha mouse liver 12.

## Data Availability

Not applicable.

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
