# Peer review of "Targeting Insulin Resistance, Reactive Oxygen Species, Inflammation, Programmed Cell Death, ER Stress, and Mitochondrial Dysfunction for the Therapeutic Prevention of Free Fatty Acid-Induced Vascular Endothelial Lipotoxicity"

_antioxidants, 2024, doi:10.3390/antiox13121486_

Round 1

Reviewer 1 Report

The authors provide a review on the therapeutic potential of targeting insulin resistance, ROS, inflammation, programmed cell death, ER stress and mitochondrial dysfunction during free fatty acid induced vascular endothelial lipotoxicity.  This is an interesting article. Some pertinent references are cited and the conclusions appear to be justified based on the information provided. The article is well structured, but can be improved.  The title of the manuscript could be modified to "Targeting  insulin resistance, reactive oxygen species, inflammation, programmed cell death, ER stress and mitochondrial dysfunction for the therapeutic prevention of free fatty acid induced vascular endothelial lipotoxicity".

1.     Introduction. The authors should provide a stronger rationale as to why this review was undertaken and a brief description of the topics to be discussed.

2. It would be very helpful if the authors can provide a figure/schemme that depicts the relationship/connection of insulin resistance, ROS, inflammation, programmed cell death, ER stress and mitochondrial dysfunction during free fatty acid induced vascular endothelial lipotoxicity in order to demonstrate why these mechanisms are being targeted.

3. The authors should describe vascular endothelial lipotoxicity in greater detail.

4.  The authors need to emphasize and elaborate on the novelty aspect of their review as well as the clinical applicability of thier findings.

5.  In my opinion, the figure presented in the conclusion section should be placed elsewhere, perhaps in the introduction section of the paper.

6. It would be nice to include a section entitled "future perspectives".

7. Line 661- others mechanism should be other mechanisms.

Author Response

  • The title of the manuscript could be modified to "Targeting insulin resistance, reactive oxygen species, inflammation, programmed cell death, ER stress and mitochondrial dysfunction for the therapeutic prevention of free fatty acid induced vascular endothelial lipotoxicity".

Ans: We rewrite the title as “Targeting insulin resistance, reactive oxygen species, inflammation, programmed cell death, ER stress and mitochondrial dysfunction for the therapeutic prevention of free fatty acid induced vascular endothelial lipotoxicity”

  • The authors should provide a stronger rationale as to why this review was undertaken and a brief description of the topics to be discussed.

Ans: We added the stronger rationale as to why this review was undertaken and a brief description of the topics to be discussed in introduction as “This review not only revealed molecular signaling such as PA could induce apoptosis through suppressed κ-opioid receptor (κ-OR). Besides, PA-induced inflammation by inhibiting the expression of Dickkopf‐1 (DDK1). PA also induces autophagy but disturbs autophagy flux by inhibiting the fusion of autophagosome and lysosome. In ROS, PA inhibited liver kinase B1 (LKB1) dependent AMPK/ Nrf-2 signaling pathway to increase intracellular ROS content. PA inhibits sarco/endoplasmic reticulum calcium ATPase (SERCA) expression and Ca2+ pump ATPase activity, which induces ER stress and activates JNK.

        Also, this review disclosed various therapeutic agents against vascular endothelial lipotoxicity through different targeting such as traditional Chinese medicine Salvianolic acid B (SAB), natural polyphenol Honokiol, natural flavonoid Dihydromyricetin (DHM), Isopropyl 3-(3,4-dihydroxy phenyl)-2-hydroxy propionate (IDHP), was used to attenuate PA-induced vascular apoptosis, pyroptosis and ferroptosis respectively. Unsaturated fatty acids like Eicosapentaenoic acid (EPA) and Docosahexaenoic acid (DHA) attenuated inflammation through reduced inflammation cytokine. Resveratrol (RSV), a natural phenol, was used to restore autophagy and improve autophagy flux disturbed by PA. Protocatechuic acid (PCA, 3,4-dihydroxybenzoic acid) is the major metabolite of polyphenols activating the LKB1/AMPK/Nrf-2 pathway to reduce ROS generation. DAPA increases mitochondrial viability, ATP production, mitochondrial biogenesis, and reduced apoptosis of PA-treated endothelial cells through the activated SIRT1/PGC-1α signaling pathway. Ilexgenin A suppresses PA-induced ER stress by activating AMPK to inhibit TNXIP/NLRP3-related IL-1b secretion. Cyanidin-3-O-glucoside (C3G) could regulate phosphorylation of p-IRS and subsequently activate the Akt/eNOS pathway to restore NO production through inhibition of IKK, and JNK is Nrf2 dependent.

       This review describes the molecular mechanism of vascular lipotoxicity caused by PA and the agents that can be used for prevention or treatment. It also provides researchers with future research directions and clinical applications in cardiovascular disease.” Line 117-143.

Ans:

  • It would be very helpful if the authors can provide a figure/schemme that depicts the relationship/connection of insulin resistance, ROS, inflammation, programmed cell death, ER stress and mitochondrial dysfunction during free fatty acid induced vascular endothelial lipotoxicity in order to demonstrate why these mechanisms are being targeted.

Ans: Figure 2 (line 299-332), figure 3 (line 432-470), figure 4 (line 654-685), figure 5 (line 747-779), figure 6 (line 820-840), figure 7 (line 896-927) are added and depicts the relationship/connection of insulin resistance, ROS, inflammation, programmed cell death, ER stress and mitochondrial dysfunction during free fatty acid induced vascular endothelial lipotoxicity.

  • The authors should describe vascular endothelial lipotoxicity in greater detail. 

Ans: We described vascular endothelial lipotoxicity in introduction as “Lipotoxicity is the accumulation of excess fatty acids in parenchymal cells of various tissues, leading to disruption of cellular homeostasis and cellular dysfunction[18]. These ectopic fat deposits may act as an active mechanism to release multiple biochemical mediators in the body that affect insulin resistance and inflammation, both of which increase cardiovascular risk[19]. Lipotoxicity affects various organs, including cardiac and vascular tissue. In the U.S., approximately 25% of adults were affected by lipotoxicity through obesity, diabetes, and heart failure[20]. Accumulated evidence demonstrated that circulating free fatty acids like palmitic acid (PA) can induce endothelial dysfunction through disruption of insulin signaling, promoting oxidative stress and inflammation, impairing mitochondrial integrity, and inducing ER stress. These different mechanisms such as PA promoted acid sphingomyelinase (ASM) which increase superoxide production via NOX2 and NOX4 to suppress IRS/PI3k/Akt/eNOS signaling pathway leading to insulin resistance of vascular endothelial in vivo and in vitro study [21]. PA-induced fragmentation of mitochondria leads to malfunction of mitochondria and loss of mitochondrial membrane potential through inhibition of tyrosyl transfer-RNA synthetase /poly (ADP-ribose) polymerase 1 (TysR/PARP1) pathway, which reduced expression of MFN1, MFN2, OPA1 leading to increase ROS[22]. PA-induced ER stress and activated unfolded protein response (UPR); meanwhile, ER stress causes endothelial mesenchymal transition (EndMT), inhibits angiogenesis (cell migration, tube number), and decreases cell viability [23].” Line 98-116.

  • The authors need to emphasize and elaborate on the novelty aspect of their review as well as the clinical applicability of their findings.

Ans: The novelty aspect of different target agent was described and added as

“This result showed that SAB may be a potential drug to treat cardiovascular disease.” (line 242-243)

“Therefore, PTX3 can be a therapeutic target of reducing atherosclerosis in clinical application.” (line 250-251)

“These findings provide a new avenue as DHM may prevent atherosclerosis induced by lipotoxicity.” (line 256-257)

“These results showed HIF1A-AS1 may be a novel molecular mechanism to against cardiovascular disease induced by hyperlipidemia.” (line 264-265)

“Several steps of this nature study were removed are limitation of this findings, thus implication of these findings on clinical inferring should be prudence” (line 275-276)

“This result provided that a therapeutic idea with IDHP for against vascular lipotoxicity-induced senescence.” (line 282-284)

“These findings revealed the critical role of unsaturated fatty acid EPA in an anti-atherogenic diet related to cardiovascular disease.” (line 376-377)

“Like EPA, DHA has an anti-atherosclerotic change in endothelial cells that prevents cardiovascular disease.” (line 382-383)

“These results disclosed that APN has a protective effect on cardiovascular disease by improving endothelial dysfunction.” (line 388-389)

“This result provided a concept that blueberry might be a food complement to lessen lipotoxicity induced vascular dysfunction” (line 391-393)

“This result showed that CCE can potentially treat inflammation and insulin resistance induced by lipotoxicity in vascular endothelial” (line 401-402)

“Therefore, homoplantaginin can be a candidate for treating or preventing vascular disease.” (line 409-410)

“These data demonstrated that apigenin can be used against vascular lipotoxicity as an alternative nutritional support.” (line 416-417)

“These results demonstrated that WA has antioxidant and anti-inflammation effects against lipotoxicity-induced endothelial dysfunction.” (line 424-426)

“MYDGF can serve as a therapeutic target for treating metabolic disorders and atherosclerosis.” (line 429-430)

“These findings implicated that small molecule glycomimics could prevent or treat of oxidative stress related disease.” (line 550-551)

“These results provided a novel antioxidant activity of phloretin through the AMPK pathway and showed it to be a food additive against vascular lipotoxicity.” (line 562-564)

“These results demonstrated PCA has promised to be a therapeutic candidate for vascular lipotoxicity, but an in vivo study could be conducted to ascertain its protective role.” (line 572-574)

“Liraglutide, therefore, has an antioxidant effect by ameliorating oxidative stress.” (line 588)

“Therefore, the extract of LBM could be supplemented to protect vascular endothelial from lipotoxicity” (line 591-592)

“These findings illuminate that DJC has endothelial protection against endothelial dysfunction under lipotoxicity” (line 597-598)

“An in vivo study could be conducted to validate pharmacokinetics and other functions of VA in the future” (line 604-605)

“GOH, therefore, has an antioxidant effect, and future studies could specifically examine its impact on vascular tone.” (line 612-613)

“These results indicated that δVB has an antioxidant effect, and SIRT3 could be developed as a novel antioxidant approach.” (line 619-620)

“These results show that CAN is a novel therapeutic medication for patients with hyperlipidemia.” (line 626-627)

“Metrnl, therefore, has a therapeutic role in defending CAD patients against endothelial dysfunction.” (line 634-635)

“This study revealed that NG-R1 has the potential to be an additional food to mitigate vascular disease and metabolic disorders.” (line 644-645)

“These implicated physcion could be a possible agent to prevent cardiovascular disease and endothelial dysfunction” (line 651-653)

“This study indicated that TBBE could be developed to counteract endothelial inflammation and vascular disease.” (line 720-721)

“These outcomes opened a new view to further study for atherosclerosis induced by fatty acid based on the TysR/PARP1 pathway” (line 726-728)

“Therefore, DAPA has the potential as a novel medication for treating hyperlipidemia” (line 733-734)

“These results provided miR-133a-3p could be developed as a potential target in preventing endothelial injury” (line 744-745)

“this study suggested ilexgenin A could be developed as an inflammatory inhibitor for antagonized free fatty acid-induced endothelial dysfunction” (line 803-805)

“These findings showed that fenofibrate can counteract macrovascular disease in the future.” (line 811-812)

“These results indicated that curcumin has a preventive effect on anti-atherosclerosis.” (line 817-818)

“This result showed that CCE has the potential to treat inflammation and insulin resistance induced by lipotoxicity in vascular endothelial” (line 861-862)

“These findings suggest that C3G can be a ROS scavenger and exert the ability to prevent insulin-resistant cardiovascular disease.” (line 869-871)

“Therefore, APNs exert cardiovascular protective effects against oxidative stress to improve endothelial dysfunction.” (line 876-877)

“These results implicated that phlorizin provided therapeutic options for patients with type II DM and vascular disease.” (line 882-883)

“The protective effect of GGLME may come from pinitol contained in it, but GGLME is also rich in flavanones and dihydrostibenes, which are helpful in metabolic disease” line (892-894)

  • In my opinion, the figure presented in the conclusion section should be placed elsewhere, perhaps in the introduction section of the paper.

Ans: We placed the figure presented in the conclusion section in the introduction. (line 144-190)

  • It would be nice to include a section entitled "future perspectives"

Ans: We added future perspectives in section 3, “From this review, we know that clinical application drugs such as dapagliflozin (DAPA),canagliflozin (CAN), and leflunomide can improve endothelial dysfunction caused by free fatty acid PA. In the future, more clinical drugs can be discovered to prevent or treat cardiovascular diseases caused by hyperlipidemia. There are relatively few epigenetic studies on endothelial dysfunction caused by PA, and research can be conducted in related aspect.” (line 1042-1047)

  • Line 661- others mechanism should be other mechanisms.

Ans: We rewrote the subtitle 2.10 as “Other mechanisms” (line 997)

Reviewer 2 Report

The authors submit a review entitled "Therapeutic prevention of targeting on insulin resistance, reactive oxygen species, inflammation, programmed cell death, ER stress and mitochondrial dysfunction during free fatty acid induced vas-4 cular endothelial lipotoxicity". As fatty acid overload induces endothelial dysfunction through various molecular mechanisms, this review summarizes the latest studies, revealing the molecular mechanism of free fatty acid-induced vascular dysfunction, targeting insulin resistance, reactive oxygen species, inflammation, programmed cell death, ER stress, and mitochondrial dysfunction.

The scientific message is interesting. The main problem is the numerous mistakes in english language; I have started correcting but there is just too much of them. This paper has to be revised before being submitted again.

There is just one figure which is limiting for a review. please take the most relevant informations of the big table, and summarize them in 2-3 figures.

Line 41-42. Rephrase "Triglycerides, phospholipids, and cholesterol esters are the three types of 41 esters that FAs usually found in organisms." Also following sentence "Free fatty acids (FFA) are circulating plasma 42 FA are not present in esters type which usually bound to albumin."

line 44 "IngestED excessive FFA"

line53"meat and meat product majorLY contributed to MUFA intake,"

and so on.

Author Response

  • The scientific message is interesting. The main problem is the numerous mistakes in english language; I have started correcting but there is just too much of them. This paper has to be revised before being submitted again.

Ans: We revised this paper carefully and checked with spelling correction.

  • There is just one figure which is limiting for a review. please take the most relevant informations of the big table, and summarize them in 2-3 figures.

Ans: Figure 2 (line 299-332), figure 3 (line 432-470), figure 4 (line 654-685), figure 5 (line 747-779), figure 6 (line 820-840), figure 7 (line 896-927) are added and depicts the relationship/connection of insulin resistance, ROS, inflammation, programmed cell death, ER stress and mitochondrial dysfunction during free fatty acid induced vascular endothelial lipotoxicity which were targeted by various agents.

  • Line 41-42. Rephrase "Triglycerides, phospholipids, and cholesterol esters are the three types of 41 esters that FAs usually found in organisms." Also following sentence "Free fatty acids (FFA) are circulating plasma 42 FA are not present in esters type which usually bound to albumin."

Ans: We rephrase the sentence as “Triglycerides, phospholipids, and cholesterol esters are FA, which exist in the body as esters. Free fatty acids (FFA) are circulating plasma FA are not present in esters type which usually bound to albumin.” (line 41-43)

  • line53"meat and meat product majorLY contributed to MUFA intake,"

Ans: We recheck the sentence as “Milk and milk products mainly contributed to SFA, meat and meat product major contributed to MUFA intake, fats and oils mainly contributed to PUFA.” at line 52-53

Round 2

Reviewer 1 Report

The authors have addressed all concerns. 

The authors have adequately revised their manuscript. I have no further comments.

Reviewer 2 Report

changes are ok

changes are ok